# R-Spondin 2 governs *Xenopus* left-right body axis formation by establishing an FGF signaling gradient

Hyeyoon Lee[1], Celine Marie Camuto[1] & Christof Niehrs ●[1,2] ✉

Establishment of the left-right (LR, sinistral, dextral) body axis in many vertebrate embryos relies on cilia-driven leftward fluid flow within an LR organizer (LRO). A cardinal question is how leftward flow triggers symmetry breakage. The chemosensation model posits that ciliary flow enriches a signaling molecule on the left side of the LRO that promotes sinistral cell fate. However, the nature of this sinistralizing signal has remained elusive. In the *Xenopus* LRO, we identified the stem cell growth factor R-Spondin 2 (Rspo2) as a symmetrically expressed, sinistralizing signal. As predicted for a flow-mediated signal, Rspo2 operates downstream of leftward flow but upstream of the asymmetrically expressed gene *dand5*. Unexpectedly, in LR patterning, Rspo2 acts as an FGF receptor antagonist: Rspo2 via its TSP1 domain binds Fgfr4 and promotes its membrane clearance by Znrf3-mediated endocytosis. Concordantly, we find that at flow-stage, FGF signaling is dextralizing and forms a gradient across the LRO, high on the dextral- and low on the sinistral side. Rspo2 gain- and loss-of function equalize this FGF signaling gradient and sinistralize and dextralize development, respectively. We propose that leftward flow of Rspo2 produces an FGF signaling gradient that governs LR-symmetry breakage.

The year 2023 marks the 25th anniversary of a landmark paper on a fundamental problem in biology, how animals establish the left-right (LR) body axis[1]. LR asymmetry of thoracic and visceral organs is a chief attribute of vertebrates. The LR axis is determined early in embryonic development and serves as the basis for the formation of all subsequent LR body asymmetries. In their 1998 study, Hirokawa and colleagues[1] addressed how LR symmetry breakage is achieved during embryogenesis. They discovered that motile cilia located in the node of early mouse embryos generate a leftward fluid flow that is essential for LR axis formation. A key prediction of their paper was that ciliary fluid flow could produce a gradient of putative morphogen along the left–right axis in the node. However, while the importance of the ciliary flow has panned out in many species besides mouse[2], a morphogen, despite intense search, has remained elusive.

LR asymmetry is regulated during embryogenesis by the LR organizer (LRO) where LR symmetry breaking takes place[2]. Defective LRO formation results in heterotaxy (HTX), ranging from malformation to misarrangement of organs across the LR axis[3–5]. Since HTX is linked to congenital heart defects and organ dysfunction of live births[5], mechanistic understanding of LR specification is also of medical interest. The LRO of fish, amphibians, and mammals harbors motile cilia, which rotate clockwise to generate a leftward fluid flow[2,6,7]. This leftward flow triggers asymmetric $Ca^{2+}$ signals[8–12] and leads to asymmetric mRNA degradation of *Dand5/Cerl2/Charon* transcripts on the left LRO margin[13–16]. *Dand5* encodes an antagonist of Nodal signaling and hence unilateral *Dand5* inhibition activates the Nodal-Pitx2 signaling cascade in the left lateral plate mesoderm (LPM), which controls organ situs during organogenesis[17,18].

While there is consensus that this LR asymmetry pathway is common to fish, amphibians, and mammals, a key question is the mechanism whereby leftward flow acts mechanistically. The two main hypotheses advanced are mechanosensation[8–10,19] and

[1]Division of Molecular Embryology, DKFZ-ZMBH Alliance, Deutsches Krebsforschungszentrum (DKFZ), 69120 Heidelberg, Germany. [2]Institute of Molecular Biology (IMB), 55128 Mainz, Germany. ✉e-mail: niehrs@dkfz-heidelberg.de

chemosensation, the asymmetric transport of a morphogen[1,20,21]. Simulations support that asymmetric transport of signaling molecules to be a robust strategy to break LR asymmetry[22]. Distribution of different fluorescently labeled proteins in mouse and rabbit LRO demonstrated that ciliary flow produces concentration gradients of extracellular proteins size-dependently between 15-50 kDa, reaching LR concentration differences of up to ~10-fold[23]. Thus, ciliary flow is able in principle to generate a concentration gradient of a signal, which promotes left-specific (sinistral) cell fates. Of note, to avoid ambiguities, in this study we adhere to the terms sinistral/dextral[24,25].

The TGFβ growth factor Nodal was suggested as a flow-mediated signal[20] but was not confirmed[26]. Notably, the symmetry breaking LR morphogen should act epistatically downstream of leftward flow but upstream of asymmetric expression of the earliest LR marker, *Dand5/Cerl2/Charon*, which has not been demonstrated for any candidate signal. Thus, the principle of a flow-mediated sinistralizing signal is well supported, but its identity remains elusive. Meanwhile, recent work in mouse and fish demonstrated that ciliary force sensing is necessary and sufficient for embryonic laterality[27,28]. So, is this the deathblow for a sinistralizing signal?

Here we introduce R-Spondin 2 (Rspo2) as a candidate in *Xenopus* embryos that fulfills three chief criteria for a sinistralizing signal: (i) an extracellular signaling protein of 28 kDa symmetrically expressed in the LRO at the time of symmetry breakage, that (ii) is sinistralizing and required for LR specification on the left side, and (iii) acts epistatically downstream of leftward flow but upstream of asymmetric gene expression of *dand5*.

R-Spondins (RSPO1-4) are secreted stem cell growth factors regulating embryonic development and stem cell maintenance[29–31]. They are well-established Wnt agonists that function by shielding Wnt receptors from ubiquitination and degradation by the E3 ubiquitin ligases ZNRF3/RNF43. RSPO2 and −3 also act as BMP antagonists[32]. We show that, surprisingly, RSPO2 is also an antagonist of Fibroblast Growth Factor receptor 4 (FGFR4). We demonstrate that in *Xenopus* at flow stages, Fgfr4 signaling is dextralizing the embryo, supporting findings in chick and rabbit of FGF as dextralizing signal[33–35]. Consistently, we find an FGF signaling gradient in the LRO, high on the right and low on the left. We propose that leftward flow of Rspo2 generates an FGF signaling gradient to promote LR-symmetry breakage.

## Results

### *Rspo2* is expressed in the LRO and is required for organ laterality and LR specification

*Xenopus* is a well-established model for LR specification, featuring evolutionary conserved mechanisms found also in mammals[36]. While exploring the expression pattern of *rspo2* in early *Xenopus* embryos, we noticed in dorso-posterior neurula explants a bilateral symmetric staining of *rspo2* in the gastrocoel roof plate, corresponding to the LRO margin. This expression is similar to that of *nodal* and the Nodal inhibitor *dand5*, key regulators of LR specification[36,37] (Fig. 1a, b). The *rspo2* marginal LRO expression domain abuts leftward flow-generating medial cells and overlaps with the domain harboring immotile cilia, that sense leftward flow[13] (Fig. 1b). *Rspo1* and *rspo3* did not show this marginal LRO expression (Supplementary Fig. 1a).

To probe into a *rspo2* function in LR asymmetry development, we overexpressed *rspo2* by injecting mRNA (low-dose to avoid axial defects[31,32]) or knocked-down *rspo2* by injecting a previously validated antisense morpholino (*rspo2* Mo)[31,32] (Fig. 1c, d). Of note, adequately controlled Mos are a widely accepted and broadly used research tool in *Xenopus* harboring large stores of maternal RNAs that escape CRISPR/Cas9-mediated genome editing[38]. Both gain- and loss-of-*rspo2* function in *Xenopus* tailbuds induced defects in organ laterality including abnormal heart looping (Fig. 1e-g) and gut looping (Fig. 1h-j), but without affecting gastrulation or main body axis formation (Supplementary Fig. 1b), supporting a role for *rspo2* in *Xenopus* LR specification.

Next, we took advantage of the early axial asymmetries in *Xenopus* that enable separate targeting of left and right embryonic lineages already at early cleavage stages[36] (Fig. 1k-p). Since the LRO predominantly derives from the dorso-marginal-zone (DMZ) of *Xenopus* embryos[36], we injected 4-cell stage embryos with *rspo2* Mo into the DMZ precursor, the left and right dorso-lateral blastomere region (Fig. 1k). While left injection prevented expression of the left-LPM marker *pitx2c*[36,39,40], right injection was without effect (Fig. 1l, m). We also examined expression of *dand5*, the earliest known asymmetrically expressed gene, which displays a characteristic anteriorly shortened streak on the left- compared to the right side in post-flow stage neurulae (Fig. 1b). *Dand5* expression became symmetric when *rspo2* Mo was injected on the left but not the right (Fig. 1n-p). We conclude that *rspo2* is expressed in the LRO and is required for LR specification upstream of *dand5*, the earliest known asymmetrically expressed gene.

### Rspo2 is a sinistralizing signal in the left-right organizer
We asked if *rspo2* functions upstream or downstream of leftward flow by conducting rescue experiments in embryos in which leftward flow was impaired (scheme; Fig. 2a, e). If Rspo2 acts upstream of leftward flow, e.g. by regulating mesoderm formation, LRO specification, or ciliogenesis, it should fail to rescue LR specification in flow-compromised embryos. Conversely, a rescue of LR specification in flow-compromised embryos would indicate that Rspo2 operates downstream of these events. We employed two methods to impair leftward flow and LR specification, injection of viscous methylcellulose (MC) into the gastrocoel of *Xenopus* neurulae, which acts mechanically[37] (Fig. 2a-d), and targeting the ciliogenic gene *gas2l2* with morpholino (*gas2l2* Mo), which impairs cilia orientation[41] (Fig. 2e-h). We employed both regimes and then injected *rspo2* mRNA on the left or the right and scored for expression of the left-LPM marker *pitx2c*. Expectedly, injection of both MC and *gas2l2* Mo inhibited *pitx2c* expression (Fig. 2b, f). Strikingly, left- but not right injection of *rspo2* mRNA robustly restored left *pitx2c* expression (Fig. 2b, f). Similarly, embryo injection with MC (Fig. 2c, d) or *gas2l2* Mo (Fig. 2g, h) induced symmetric *dand5* expression at the LRO and this was once again substantially rescued by left injection of *rspo2* mRNA. The results indicate that Rspo2 acts downstream of leftward flow.

We conclude that *rspo2* is symmetrically expressed in the LRO at the right time and place for the LR signal, it is necessary for left LPM specification, and it acts upstream of *dand5* but downstream of leftward flow, as postulated for a flow-transported sinistralizing morphogen (Fig. 2i).

### Rspo2 regulates LR asymmetry Wnt and BMP independently
Rspo2 acts both as a Wnt agonist and BMP signaling antagonist during early *Xenopus* axial specification[31,32]. Since Wnt/β-catenin signaling is required for the earliest step of LR asymmetry development by specifying the LRO precursor during gastrulation[42,43] and BMP signaling may also regulate the leftward Nodal cascade[44,45], it was essential to dissect its mechanism of action. We took advantage of the modular composition of RSPO2, its FU1 domain binding to ZNRF3/RNF43[46,47], FU to LGRs[48,49], and TSP1 to BMPR1A[32]. Single Rspo2 domain mutants can be used to distinguish between Wnt and BMP signaling in *Xenopus*[31,32]. Hence, we injected *rspo2* mRNA wild-type (WT) and FU1, FU2, and TSP1 mutants and scored for *pitx2c* expression (Supplementary Fig. 1c-f). Importantly, *rspo2* mRNA induced bilateral *pitx2c* expression and hence could override right LPM specification. However, none of the single domain mutants did (Supplementary Fig. 1f). The fact that all three Rspo2 domains were essential indicates that either both Wnt agonism and BMP antagonism are simultaneously required in sinistralizing signaling, or that an altogether different signaling mode is involved. To address BMP and Wnt signaling and to further exclude the confounding possibility of Rspo2 acting during early axis formation or mesoderm

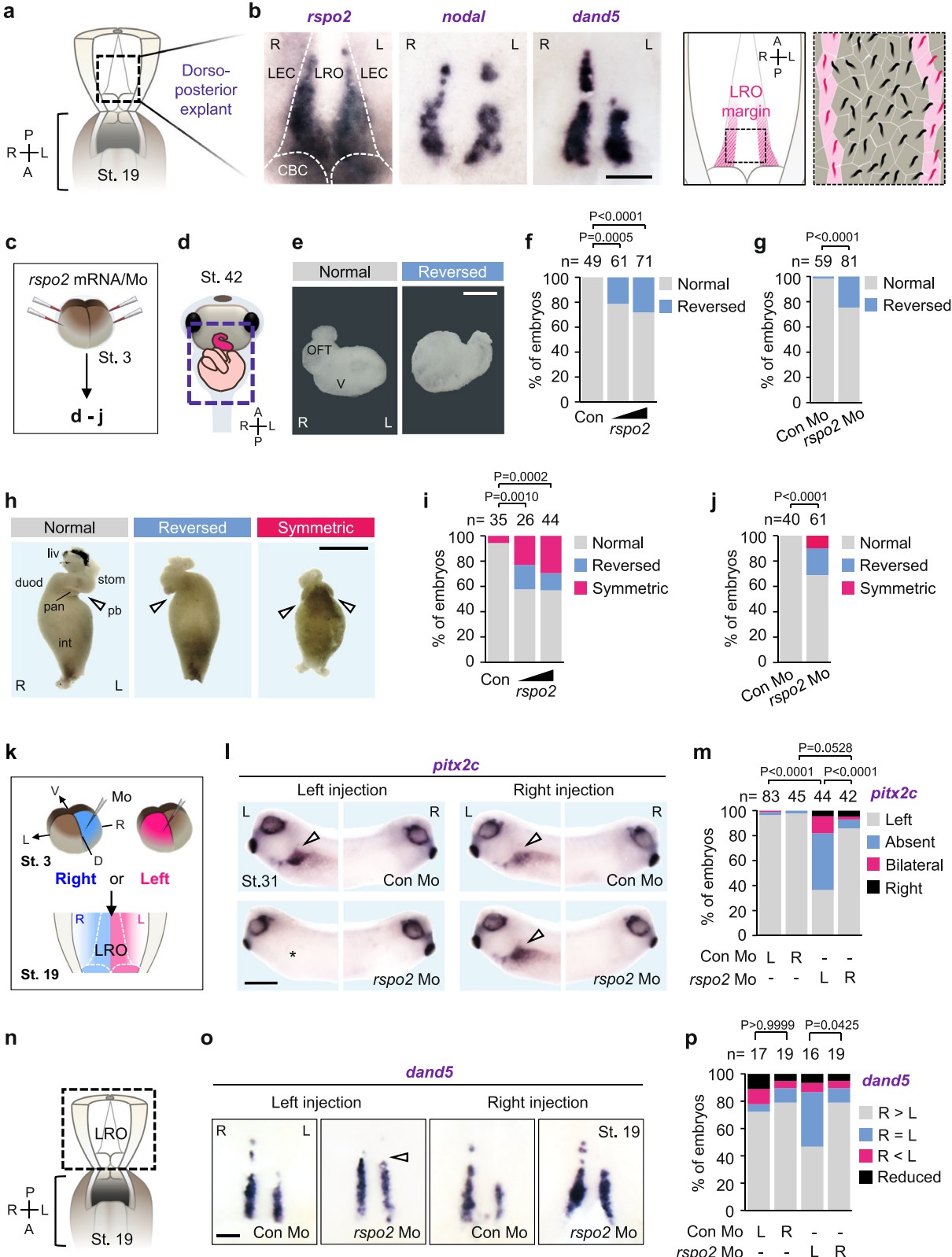

specification, we conducted gastrocoel injections at the onset of leftward flow (St. 15) (Fig. 3a). Injection of recombinant human RSPO2 protein robustly induced *pitx2c* in the right LPM (Fig. 3b, c), corroborating Rspo2 as a sinistralizing signal that can override right LPM specification also at flow stages. Importantly, neither injection of Wnt3A nor the Wnt antagonist DKK1 proteins affected *pitx2c* expression (Fig. 3b, c), despite both being active (Supplementary

Fig. 1g, h). Moreover, injection of dominant-negative *bmpr1a* (*bmpr1a^DN*) mRNA also had no effect on *pitx2c* expression (Supplementary Fig. 1i), despite strong inhibition of the BMP target gene *sizzled* expression (Supplementary Fig. 1j). Additionally, *bmpr1a^DN* failed to rescue impaired *pitx2c* expression in *rspo2* Morphants (Supplementary Fig. 1k). Taken together, we conclude that *rspo2* regulates LR asymmetry Wnt- and BMP-independently.

**Fig. 1 | Rspo2 is expressed in the LRO and is required for LR specification.**
**a** Scheme of dorso-posterior explants from *Xenopus* St. 19 neurula harboring the
left-right organizer (LRO). A anterior, P posterior, R right and L left. **b** (Left) Whole-
mount in situ hybridization (WISH) of *rspo2, nodal* and *dand5* in the LRO. LEC,
lateral-endodermal crest; CBC, circumblastoporal collar. Scale bar, 100 μm. (Right)
Cartoons illustrating the ciliated LRO. Central motile cilia (black) generate leftward
fluid flow that is sensed by immotile cilia (magenta). **c** Microinjection strategy for
(**d**–**j**). **d** Scheme of the morphologic analysis. Dashed square, ventral view of heart
and intestine from *Xenopus* tailbud (St. 42) presented in (**e**–**j**). **e** Representative
images of normal and reversed heart in *Xenopus* tailbuds. OFT outflow tract, V
ventricle. Scale bar, 100 μm. R right, L left. **f, g** Quantification of heart looping in
*Xenopus* tailbuds injected as indicated. **h** Representative images of normal and LR
defective *Xenopus* gut. Gut looping defects were categorized by position of the
pancreatic bay concavity in developing midgut loops. Normal showed left-sided

bay. Reversed showed right-sided bay. Symmetric showed concavities on both
sides. Arrowheads, pancreatic bay concavities. liv, liver; duod, duodenum; stom,
stomach; pan, pancreas; int, intestine. Scale bar, 1 mm. **i, j** Quantification of gut
looping in *Xenopus* tailbuds injected as indicated. **k** Left-right-specific microinjec-
tion strategy for (**l**–**p**). D dorsal, V ventral, L left and R right. **l** WISH of *pitx2c* in
*Xenopus* tailbuds (St. 31). Left (L) and right (R) sides of the same embryos are shown.
Arrowheads, *pitx2c* at the lateral plate mesoderm (LPM); Asterisk, absent *pitx2c* at
the left-LPM. Scale bar, 0.5 mm. **m** Quantification of (**l**). L left-DMZ injection, R right-
DMZ injection. **n** Scheme of dorso-posterior explants. A anterior, P posterior, R
right, L left. **o** WISH of *dand5* in *Xenopus* LRO (St. 19) injected as indicated.
Arrowhead, derepression of *dand5*. Scale bar, 100 μm. **p** Quantification of (**o**). Data
information: Two-sided Fisher's exact test used for all statistical analyses. *n* =
number of analyzed embryos. Source data are provided as a Source Data file.

## Fgfr4 is a dextralizing signal in the left-right organizer

If neither Wnt agonism nor BMP antagonism is involved, how then
Rspo2 acts as a sinistralizing signal? *Xenopus* FGF receptor 4 (Fgfr4) is a
potential heterotaxy gene expressed in LRO precursors and necessary
for *Xenopus* LR specification[50]. Given that RSPOs function as receptor
endocytosers, and Rspo2 is implicated in FGF inhibition[51], we con-
sidered that Rspo2 antagonizes Fgfr4 signaling. Since understanding
the role of FGF signaling is complicated by the sequential roles of this
pathway in LR specification before leftward flow stages[50,52,53], we
characterized Fgfr4 signaling specifically during flow stages. To bypass
early development, we inhibited Fgfr4 using gastrocoel injection of the
specific FGFR4 inhibitor BLU9931[54] (Fig. 3d), which induced robust
bilateral *pix2c* expression (Fig. 3e, f). This important result indicates
that different from early stage inhibition[50], at leftward flow stage,
Fgfr4-like signaling is essential for repressing left-LPM fate on the right.
Conversely, we conditionally activated Fgfr4 signaling by injecting
mRNA encoding a chimeric Fgfr4 (*ifgfr4*)[55], whose signaling is inducible
by the dimerizer AP20187. Injection of AP20187 into the gastrocoel at
leftward flow stages inhibited *pitx2c* expression, i.e. active Fgfr4 pro-
motes dextral specification (Fig. 3g–i and Supplementary Fig. 1l). Thus,
during *Xenopus* leftward flow stages, Fgfr4 signaling acts as a dex-
tralizing signal that represses left LPM specification, consistent with
findings in chick and rabbit of FGF8[33–35,56].

To examine Rspo2 as an Fgfr4 inhibitor in LR signaling, we tested
if forced Fgfr4 signaling rescues Rspo2-induced LR defects (scheme;
Supplementary Fig. 1m). Indeed, activating *ifgfr4* rescued the *rspo2*
mRNA-induced decrease of *dand5* expression (Fig. 3j, k). However, an
analogous experiment with an inducible Fgfr1 construct (*ifgfr1*) (Sup-
plementary Fig. 1n) showed no effect of *dand5* expression (Fig. 3k),
indicating receptor specificity. Conversely, loss of *pitx2c* expression in
*rspo2* Morphants was rescued by inhibiting Fgfr4 signaling via BLU9931
gastrocoel injections (scheme; Supplementary Fig. 1o and Fig. 3l). The
results indicate that Rspo2 is not only necessary but also sufficient to
sinistralize the embryo. They support that dextralizing FGF- and
sinistralizing anti-FGF signaling by Rspo2 control LR symmetry
breakage (Fig. 3m).

## RSPO2 is an FGFR4 antagonist

To further unravel its mode of action, we asked whether RSPO2 blocks
signaling by the FGF19 subfamily (FGF19, FGF21 and FGF23), the pre-
ferred FGFR4 ligands[57]. As a readout, we monitored phosphorylated
ERK1/2 (pERK1/2), a well characterized marker for the activated FGF-
Ras-MAPK signaling cascade[51,58]. In RSPO2-responsive[32] human hepa-
tocellular carcinoma HEPG2 cells, among RSPO1-4, only RSPO2
decreased FGF19-mediated ERK phosphorylation (Supplementary
Fig. 2a, b), while all RSPOs similarly potentiated Wnt signaling (Sup-
plementary Fig. 2c). In contrast, FGF21 and FGF23 signaling remained
unaffected by RSPO1-4 (Supplementary Fig. 2d–g). Similarly, in *Xeno-
pus* animal cap explants, injection of *rspo2* mRNA repressed

FGF19 signaling (Supplementary Fig. 2h). We analyzed RSPO2 domain
mutants in FGF reporter inhibition assays and found as in LR devel-
opment, that all three domains, FU1, FU2, and TSP1, were required
(Supplementary Fig. 2i).

Corroborating Wnt- and BMP-independent Rspo2 function in LR
specification (Fig. 3 and Supplementary Fig. 1), FGF19 signaling inhi-
bition by RSPO2 was unaffected by si*β-catenin*, si*LRP5/6* and si*BMPR1A*
(Fig. 4a, b and Supplementary Fig. 2j, k). Likewise, treatment with
recombinant Wnt3A and BMP antagonist Noggin, or BMPR inhibitor
LDN193189 had no effect on FGF19 signaling (Supplementary
Fig. 2l–n). In contrast, si*LGR4/5* completely rescued FGF19 inhibition by
RSPO2 (Fig. 4c), consistent with the requirement for the LGR-binding
FU2 domain in RSPO2-mediated FGF19 signaling inhibition (Supple-
mentary Fig. 2i).

To investigate the requirement for RSPO2 to inhibit FGF signaling,
we analyzed *Xenopus* neurulae, where injection of *rspo2* Mo increased
pERK levels, irrespective of *lrp6* Mo and *bmpr1a^DN^* (Fig. 4d). Likewise, in
H1581 cells, which express endogenous *RSPO2*[32], si*RSPO2* but not
si*LRP5/6* RNA sensitized cells to FGF19 stimulation, increasing pERK
levels (Fig. 4e, f). In contrast, BMPR1A overexpression had no effect on
FGF19 signaling (Fig. 4g, h). We confirmed that FGF19 signaling
occurred via FGFR4, since BLU9931 eliminated si*RSPO2*-mediated
upregulation of FGF19 signaling (Fig. 4i). Taken together, we conclude
that RSPO2 inhibits FGF19-FGFR4 signaling independently of Wnt
and BMP.

## RSPO2 binds FGFR4 via the TSP1 domain

RSPOs are receptor endocytosers[59], suggesting that RSPO2 may
interact directly with FGFR4. Hence, we carried out cell surface binding
assays, a sensitive and quantitative method to monitor RSPO ligand-
receptor interactions[32]. We tested binding of alkaline-phosphatase
(AP)-tagged RSPO1-4 fusion proteins to cells transfected either with
FGFR4 or with LGR4 as a positive control (Fig. 5a, b and Supplementary
Fig. 3a, b). Expectedly, all four RSPOs bound to LGR4 (Supplementary
Fig. 3a, b). Interestingly, FGFR4 bound strongly to RSPO2, weakly to
RSPO3, and not to RSPO1 and RSPO4 (Fig. 5b). Unlike FGFR4, FGFR1
bound only weakly to RSPO2/3 (Supplementary Fig. 3c). Titration
experiments with RSPO2 and the extracellular domain (ECD) of FGFR4
confirmed high affinity binding ($K_d = 1.5$ nM) (Fig. 5c), comparable to
previously described RSPO-receptor interactions[32,60]. To delineate the
domains required for FGFR4 binding, we tested RSPO2 deletion
mutants in cell surface–(Fig. 5d, e and Supplementary Fig. 3d, e) and in
in vitro binding assays (Supplementary Fig. 3f). Both assays showed
that FGFR4 binding required the TSP1–but not the FU1/2 domains of
RSPO2. A TSP1 domain swap between RSPO2 and RSPO1 (chimera R1-
TSPR2)[32] conferred FGFR4 binding also to RSPO1, corroborating the
importance of the TSP1 domain (Supplementary Fig. 3g–i). We con-
clude that RSPO2 is a high affinity ligand of FGFR4 and interacts via its
TSP1 domain.

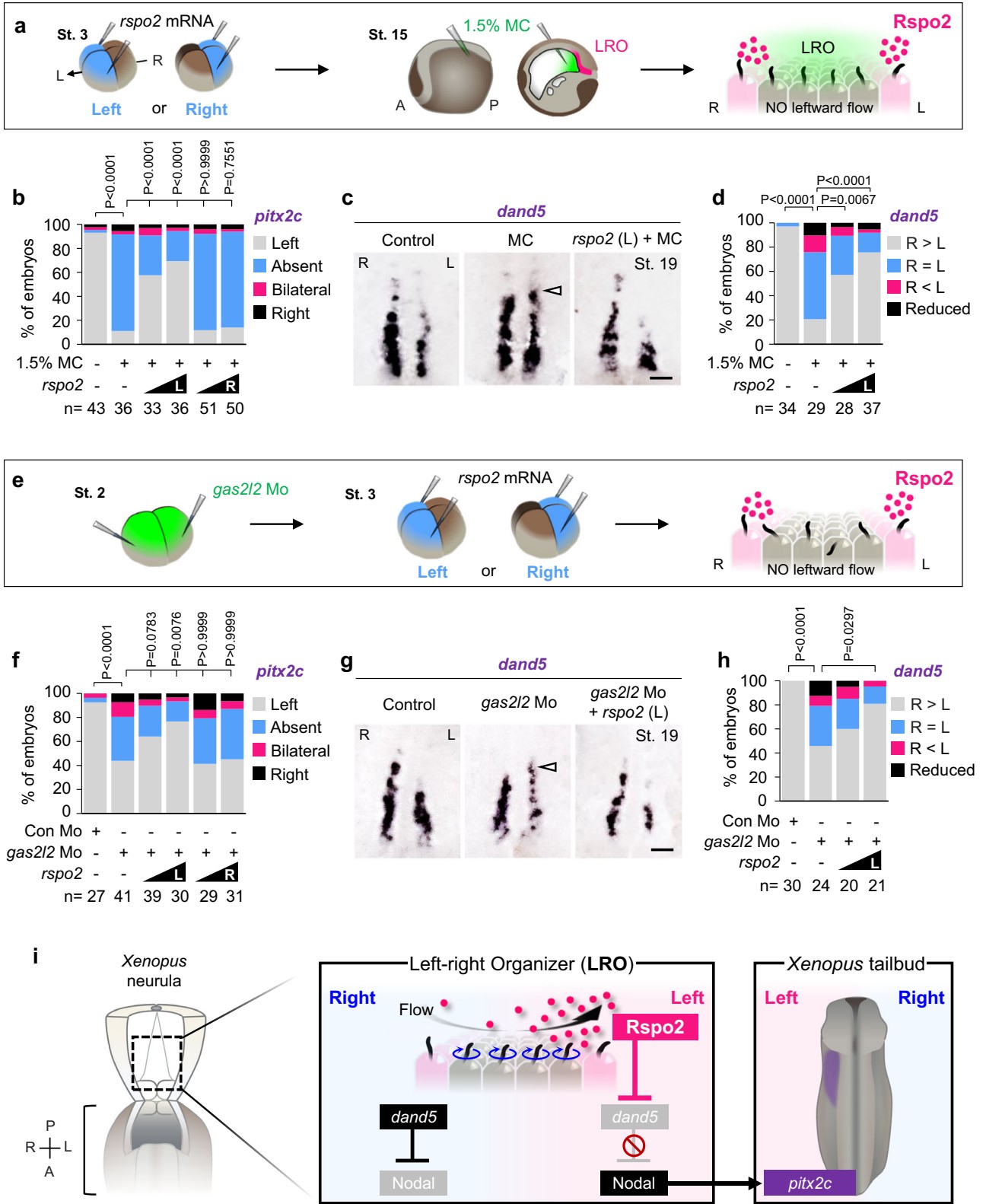

**Fig. 2 | Rspo2 is a sinistralizing signal in the left-right organizer. a** Microinjection strategy for (**b**–**d**). MC, methylcellulose. **b** Quantification of *pitx2* in *Xenopus* tail-buds (St. 31) injected as indicated. L left-blastomeres injection, R right-blastomeres injection. *n* = number of embryos. **c** WISH of *dand5* in *Xenopus* LRO (St. 19). Arrowheads, *dand5* derepression. Scale bar, 100 μm. **d** Quantification of (**c**). *n* = number of dorso-posterior explants. **e** Microinjection strategy for (**f**–**h**). **f** Quantification of *pitx2* in *Xenopus* tailbuds (St. 31) injected as indicated. L left-

blastomeres injection, R right-blastomeres injection. *n* = number of embryos. **g** WISH of *dand5* in *Xenopus* LRO (St. 19). Arrowheads, *dand5* derepression. Scale bar, 100 μm. **h** Quantification of (**g**). *n* = number of dorso-posterior explants. **i** Model for Rspo2 function as a leftward signal in *Xenopus* LRO. Data information: Two-sided Fisher's exact test used for all statistical analyses. Source data are provided as a Source Data file.

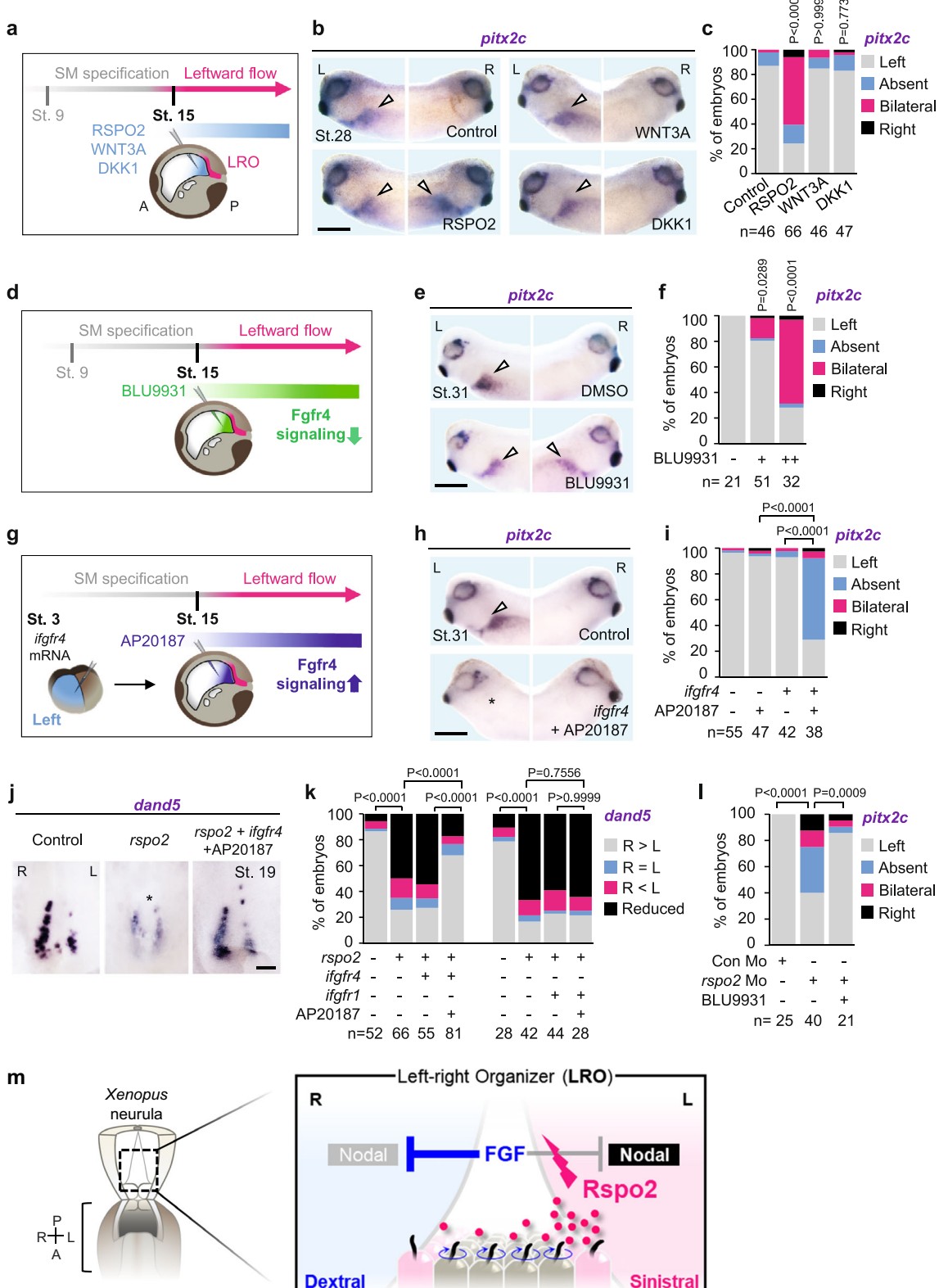

## RSPO2 triggers FGFR4 clathrin-mediated endocytosis and engages ZNRF3

The common mode of action of RSPOs in Wnt and BMPR1A signaling is that of a multivalent adapter molecule to ZNRF3/RNF43 E3 ligases, which couples to- and mediates endocytosis and cell surface removal of target transmembrane proteins. To test if RSPO2 also removes cell surface FGFR4, we carried out cell surface biotinylation assays, which

confirmed that RSPO2 greatly reduces cell surface FGFR4 levels (Supplementary Fig. 4a, b). Once again, this effect was unperturbed by si*LRP5/6* (Supplementary Fig. 4a) and si*BMPR1A* (Supplementary Fig. 4b) but not si*LGR4/5* (Supplementary Fig. 4a). Conversely, si*RSPO2* treatment of H1581 cells stabilized cell surface FGFR4, while si*LRP5/6* showed no effect (Supplementary Fig. 4c). Immunofluorescence microscopy (IF) showed that in H1581 cells, FGFR4 and FGFR1 localize

**Fig. 3 | Rspo2 regulates LR asymmetry by antagonizing Fgfr4 in the LRO.**
**a** Gastrocoel microinjection strategy for (**b**, **c**). Schematic representation of *Xenopus* neurula (St. 15) is shown in sagittal section indicating the LRO (magenta). SM, superficial mesoderm. **b** WISH of *pitx2c* in *Xenopus* tailbuds (St. 31) injected with recombinant proteins. Left (L) and right (R) sides of the same embryos are shown. Arrowheads, *pitx2c* at the LPM. Scale bar, 0.5 mm. **c** Quantification of (**b**). *n* = number of embryos. **d** Gastrocoel microinjection strategy for (**e**, **f**). **e** WISH of *pitx2c* in *Xenopus* tailbuds (St. 31) injected with BLU9931. Arrowheads, *pitx2c* expression at the LPM. Scale bar, 0.5 mm. **f** Quantification of (**e**). *n* = number of embryos. **g** Microinjection strategy for (**h**, **i**). **h** WISH of *pitx2c* in *Xenopus* tailbuds (St. 31) with

AP20187-mediated induction of Fgfr4 signaling. Arrowheads, *pitx2c* at the LPM. Asterisk, absent *pitx2c* at the left-LPM. Scale bar, 0.5 mm. **i** Quantification of (**h**). *n* = number of embryos. **j** WISH of *dand5* in *Xenopus* LRO (St. 19) injected as indicated. Asterisk, reduced *dand5*. Scale bar, 100 μm. **k** Quantification of *dand5* in *Xenopus* LRO (St. 19) with induction of Fgfr4 signaling (**j**) and Fgfr1 signaling along with *rspo2* mRNA injection. *n* = number of dorso-posterior explants. **l** Quantification of *pitx2* in *Xenopus* tailbuds (St. 31) injected as indicated. *n* = number of embryos. **m** Anti-FGF signaling by Rspo2 controls LR-symmetry breakage in the LRO. Data information: Two-sided Fisher's exact test used for all statistical analyses. Source data are provided as a Source Data file.

in cytoplasmic vesicles (Supplementary Fig. 4d-g). Importantly, si*R-SPO2* and si*LGR4/5* but not si*LRP5/6* treatment shifted FGFR4 (Supplementary Fig. 4d, e) but not FGFR1 to the cell surface (Supplementary Fig. 4f, g), corroborating that RSPO2 targets FGFR4 but not FGFR1 (Fig. 3k and Supplementary Fig. 3c).

To further investigate targeted FGFR4 internalization, we carried out endocytosis assays. Cell surface proteins were labeled extracellularly with a biotin-coupled cross-linker containing a disulfide bridge, which can be cleaved-off by the reducing agent MesNa. At $t_0$, extracellular domain labeled FGFR4 was quantitatively removed by MesNa (Fig. 5f), i.e. no FGFR4 was internalized. At $t_{30\ min}$, only a faint band of MesNa-protected FGFR4 appeared, indicating little endocytosis during this interval. In contrast, RSPO2 treatment led to robust MesNa-protection of FGFR4, indicative of enhanced internalization (Fig. 5f).

Consistently, colocalization of FGFR4 with the early endosome marker EEA1 (Fig. 5g, h) and clathrin (Fig. 5i, j) was greatly increased by RSPO2 treatment. Conversely, FGFR4-EEA1 colocalization was reduced by si*RSPO*2 (Fig. 5k, l). However, FGFR4 hardly colocalized with the lysosomal marker Lamp1 upon RSPO2 treatment (Supplementary Fig. 4h, i), indicating that RSPO2 does not induce lysosomal targeting. We conclude that RSPO2 binds FGFR4, induces its clathrin-mediated endocytosis, and thereby desensitizes cells to FGF19 stimulation.

Since RSPOs engage the transmembrane E3 ubiquitin ligase ZNRF3 and its homolog RNF43 to induce endocytosis of their receptor targets, we envisaged that ZNRF3/RNF43 are also important to antagonize FGFR4 signaling. Indeed, in H1581 cells, si*ZNRF3/RNF43* treatment potentiated FGF19 stimulation (Supplementary Fig. 4j), indicative of FGFR4 derepression. Cell surface biotinylation assays showed that RSPO2 requires ZNRF3/RNF43 to eliminate cell surface FGFR4 (Fig. 5m) and IF confirmed RSPO2, FGFR4 and ZNRF3 colocalization (Fig. 5n). Taken together, our results support a model (Fig. 5o) whereby (1) RSPO2 forms a quaternary complex with ZNRF3, LGRs, and FGFR4 via its FU1, FU2 and TSP1 domains, respectively, (2) promotes membrane clearance of FGFR4 via clathrin-mediated endocytosis, and thereby (3) desensitizes cells to FGF signaling.

We analyzed if this RSPO2 mode of action in human cells is conserved in *Xenopus*. In dorsoposterior explants, *fgfr4* was expressed in the LRO margin as well as the lateral-endodermal crest (LEC), while *fgfr1* expression was restricted to the LRO margin. *Znrf3*, *lgr4*, and *fgfr19* were coexpressed in the LRO margin (Supplementary Fig. 5a). Cell surface binding assays showed binding of recombinant *Xenopus* Rspo2 to Fgfr4 via the TSP1 domain (Supplementary Fig. 5b-d), similar to human RSPO2-FGFR4 binding (Fig. 5e). To demonstrate that Rspo2 also eliminates cell surface Fgfr4 in *Xenopus*, we analyzed animal cap explants that were mRNA-coinjected with *fgfr4*-EYFP to follow the receptor, *membrane-bound*-RFP to label plasma membranes, and either *rspo2* wildtype to induce receptor endocytosis, or the TSP1 deletion mutant (*rspo2*$^{ΔTSP}$) lacking the Fgfr4 binding site. In control and *rspo2*$^{ΔTSP}$ injected explants, Fgfr4-EYFP colocalized with RFP in the plasma membrane (Supplementary Fig. 5e-g). In contrast, Fgfr4-EYFP levels were greatly

reduced by *rspo2*, suggesting that in *Xenopus*, different from human cells, internalized Fgfr4 is not only internalized but also degraded. This *rspo2*-induced reduction of cell surface Fgfr4 was prevented by dominant-negative *znrf3* (*znrf3*$^{ΔR}$)[61] (Supplementary Fig. 5h-j), confirming the importance of *znrf3* to eliminate Fgfr4. The requirement of *znrf3* in LR specification was corroborated in *znrf3* morphants (Fig. 5p, q), where injection of previously validated *znrf3* Mo[32,62] prevented *pitx2c* expression at the left-LPM. Misexpression of *pitx2c* in the morphants was rescued by co-injection of Mo-resistant human ZNRF3 mRNA[62], demonstrating specificity of the Mo. Collectively, we conclude that Rspo2 binds and promotes membrane clearance of Fgfr4 in a Znrf3-dependent manner, analogous to the mechanism in human cells.

## RSPO2$^{TSP1}$-derived peptides impair RSPO2-FGFR4 binding and LR specification

The RSPO2-TSP1 domain is only 65 amino acid small, yet it mediates not only FGFR4- but also BMPR1A binding[32,63]. Hence, the question arose if FGFR4 and BMPR1A sterically compete for RSPO2 binding. Remarkably, the TSP1 binding sites for both receptors appeared to be distinct: In cell surface binding assays, recombinant FGFR4$^{ECD}$ completely abolished RSPO2 binding to FGFR4 but had no effect towards BMPR1A binding (Fig. 6a-c). Conversely, BMPR1A$^{ECD}$ competed RSPO2 binding to BMPR1A, while it had no effect on RSPO2 binding to FGFR4. This result indicated a distinct TSP1 binding site for FGFR4 and raised the possibility to design specific inhibitors.

To narrow down the FGFR4 binding site, we screened overlapping 10-mer peptides spanning the TSP1 domain (Supplementary Fig. 5a) for their ability to competitively disrupt RSPO2 binding to FGFR4 but not to BMPR1A[63] (Supplementary Fig. 5b, c). We identified two overlapping peptides, TK (TRQIV**KKPVK**) and KC (**KKPVK**DTILC) that at 100 μM markedly reduced the RSPO2-FGFR4$^{ECD}$ interaction (Supplementary Fig. 5c). Importantly, peptide RW (RNNRTSGFKW), which prevents RSPO2-BMPR1A$^{ECD}$ interaction[63], showed no effect on RSPO2-FGFR4$^{ECD}$ binding (Supplementary Fig. 5c). Cell surface binding assays confirmed that TK and KC, but not RW prevent RSPO2-FGFR4$^{ECD}$ interaction, without affecting RSPO2-BMPR1A$^{ECD}$ interaction (Fig. 6d, e). Interestingly, the TK and KC motifs of RSPO2 are highly conserved between species but not conserved in RSPO1, 3, and 4 (Fig. 6f, g), suggesting that this site specifically evolved for high-affinity FGFR4 binding.

We leveraged the specificity of TK and KC peptides to dissociate multimodal Rspo2 signaling and to selectively disrupt the Rspo2-Fgfr4 antagonism in *Xenopus* LR development. Injection of TK and KC, but not RW into the gastrocoel of leftward flow stage neurulae (St. 15) greatly increased FGF/ERK signaling in LRO explants (St. 18) without affecting pSmad1/BMP signaling (Fig. 6h, i) or Wnt signaling (Supplementary Fig. 6d). Remarkably, *Xenopus* tailbuds injected with TK into the gastrocoel of neurulae at leftward flow stage showed symmetric expression of *dand5* (Fig. 6j, k). We conclude (1) that the TSP1 domain of RSPO2 harbors binding sites for FGFR4 that are distinct from those for BMPR1A, which (2) engage the TK-KC peptide-spanning sequences of RSPO2$^{TSP1}$, that (3) can be used to specifically interfere with the ability of Rspo2 to inhibit Fgfr4 signaling (Fig. 6l), corroborating Rspo2 as a sinistralizing signal in *Xenopus*.

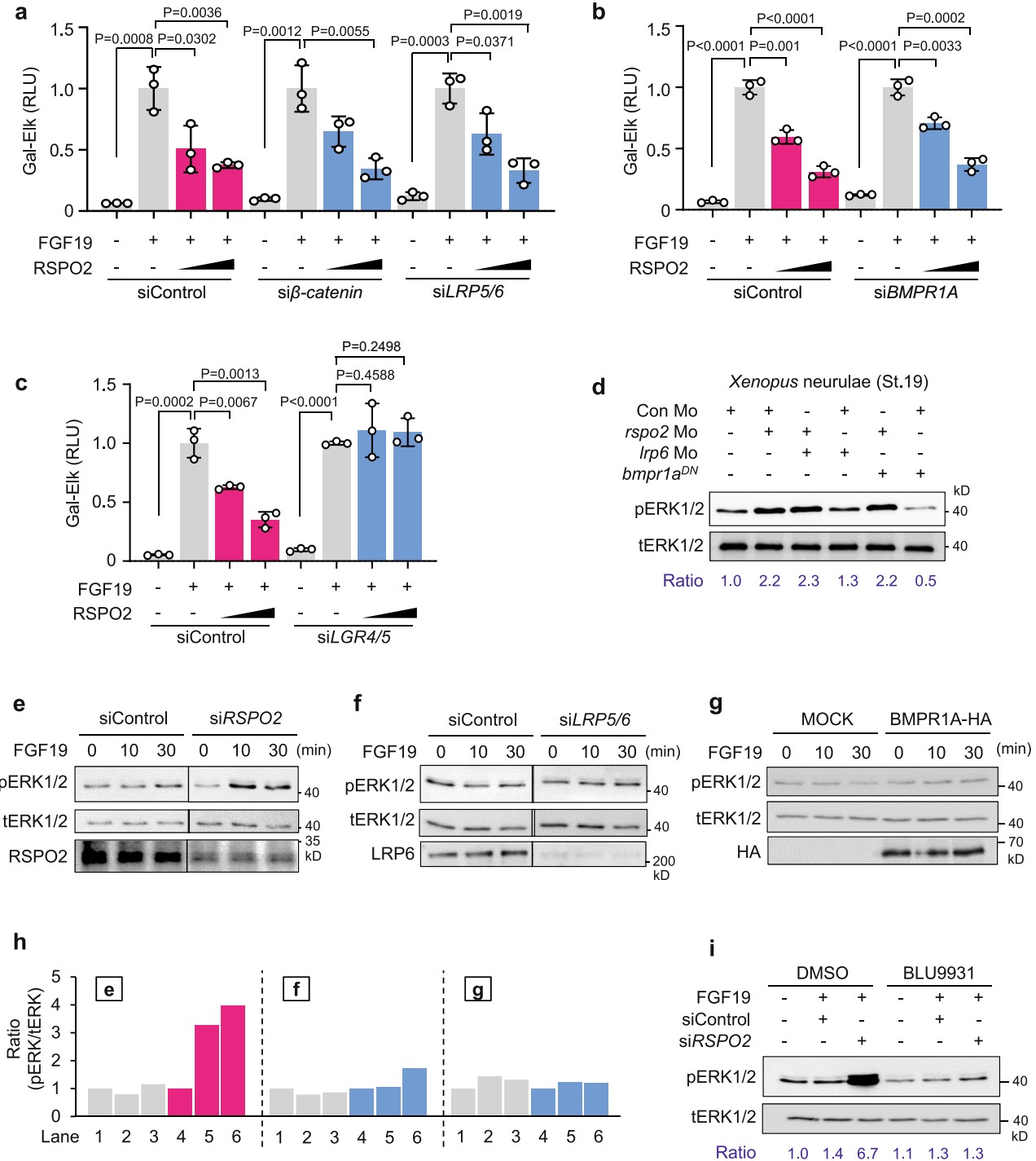

**Fig. 4 | RSPO2 antagonizes FGF19-FGFR4 signaling Wnt and BMP independently.** **a**, **b** FGF-responsive GAL-Elk1 luciferase reporter assays in HEPG2 cells treated as indicated. **c** GAL-Elk1 luciferase reporter assays in HEPG2 cells treated as indicated. **d** Western blot analysis in *Xenopus* neurulae (St. 19) injected as indicated. **e**–**g** Western blot analyses in H1581 cells treated with siRNA and stimulated by FGF19 as indicated. **h** Quantification of (**e**–**g**). Ratio, relative levels of pERK1/2 normalized to tERK1/2. **i** Western blot analysis in H1581 cells treated as indicated. Ratio, relative levels of pERK1/2 normalized to tERK1/2. Data information: For all reporter assays, data are displayed as mean ± SD with two-tailed unpaired *t*-test. *n* = 3 biologically independent samples. Source data are provided as a Source Data file.

## Rspo2 generates an FGF signaling gradient in the left-right organizer

A key prediction of our model (Fig. 3m) is that Rspo2 by way of its Fgfr4 antagonism, induces an FGF signaling asymmetry in the LRO, with right – and left sides exhibiting high and low FGF signaling, respectively. To investigate this possibility, we performed IF in pre-flow and flow stage LRO explants for pERK. No asymmetric pERK1 staining was detected in pre-flow LRO (St. 14) (Fig. 7a, b and Supplementary Fig. 7a). Strikingly, at flow-stage (St. 18), right side LRO showed robustly higher pERK1 levels than the left side (Fig. 7c and Supplementary Fig. 7b). The fluorescence intensity profile of pERK1 across the LR axis indicated an FGF signaling gradient with a 6-fold slope between right and left side (Fig. 7d). Gastrocoel injection of recombinant human RSPO2 protein drastically reduced- and

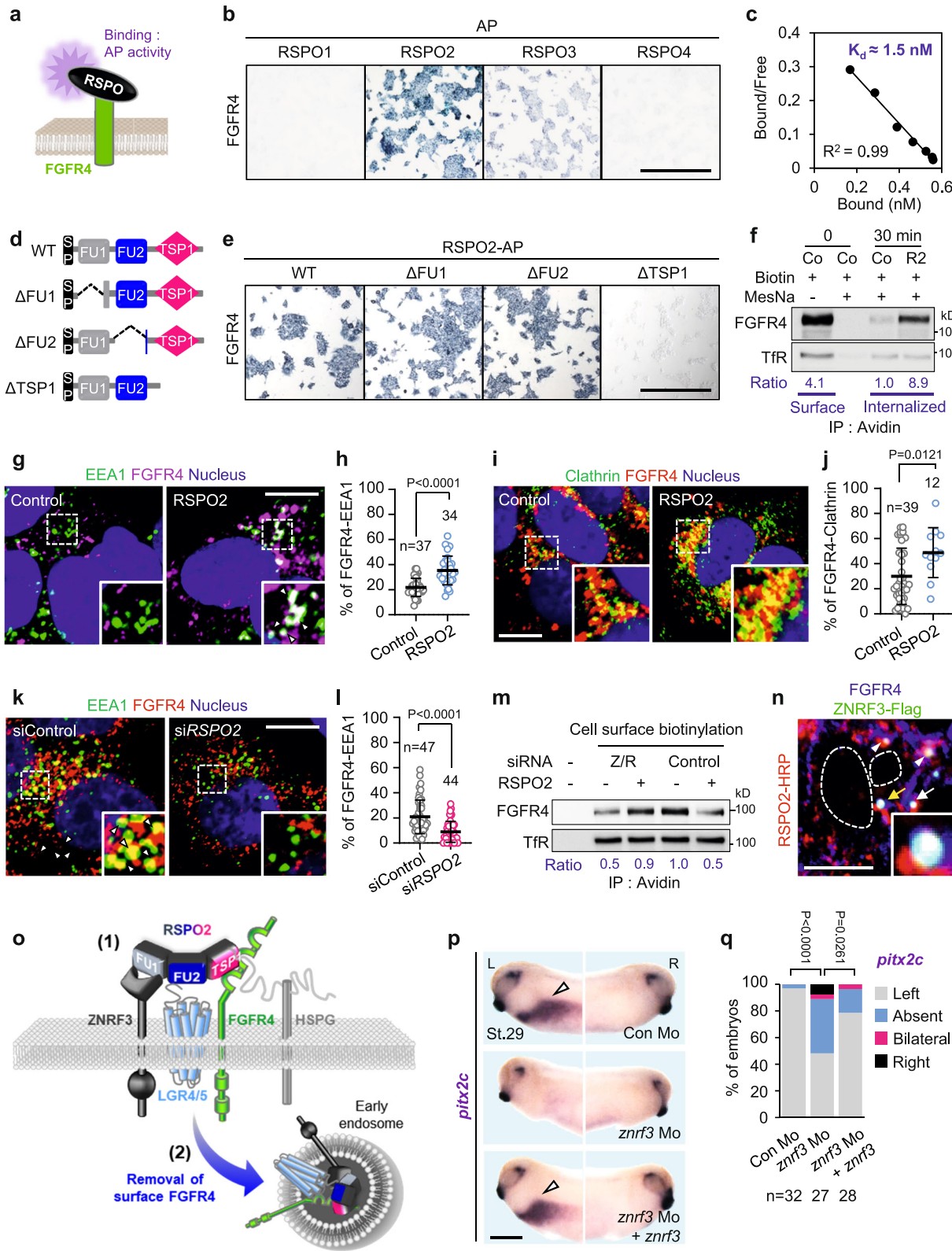

equalized the pERK1 gradient in the LRO (Fig. 7c, d and Supplementary Fig. 7b) similar to FGFR4 inhibitor BLU9931 (Fig. 7e, f and Supplementary Fig. 7c), corroborating Rspo2 as FGF signaling antagonist. Conversely, injection of *rspo2* Mo increased and equalized the pERK1 gradient (Fig. 7g, h and Supplementary Fig. 7d) similar to gastrocoel injection of methylcellulose (MC), which impairs leftward flow (Fig. 7e, f and Supplementary Fig. 7c).

To confirm these results, we bisected flow-stage LRO explants and separately analyzed left and right halves by western blot for pERK1/2. Once again, right side LRO halves showed ~3-fold higher pERK1/2 levels than left halves (Supplementary Fig. 7e–g). Note that these explants also contain neuroectodermal and epidermal cells not subject to flow regulation, hence the result underestimates the true LR difference of pERK1/2 in the LRO. Expectedly, RSPO2 gastrocoel injection reduced

**Fig. 5 | RSPO2 binds and eliminates cell surface FGFR4 by endocytosis.**
**a** Scheme of RSPOs cell surface binding assays in (**b**, **e**). **b** Representative images of HEK293T cells transfected and treated as indicated. Scale bar, 1 mm. **c** Scatchard analysis of RSPO2 and FGFR4ECD binding. **d** Domain structures of RSPO2 and deletion mutants. SP signal peptide, FU furin domain, TSP1 thrombospondin domain 1. **e** Representative images of HEK293T cells transfected and treated as indicated. Scale bar, 1 mm. **f** Receptor internalization assay in HEPG2 cells treated with control (Co) or RSPO2 medium (R2) as indicated. TfR, transferrin receptor. (**g**) Co-immunofluorescence microscopy (Co-IF) for EEA1 and FGFR4 in HEPG2 cells. Nuclei were stained with Hoechst. White arrowheads, colocalized FGFR4-EEA1 in magnified inset. Scale bar, 20 μm. (**h**) Quantification of (**g**). **i** Co-IF for clathrin and FGFR4 in HEPG2 cells. Nuclei were stained with Hoechst. Scale bar, 20 μm. **j** Quantification of (**i**). **k** Co-IF for EEA1 and FGFR4 in H1581 cells. Nuclei were stained with Hoechst.

White arrowheads, colocalized FGFR4-EEA1 in magnified inset. Scale bar, 20 μm. **l** Quantification of (**k**). **m** Cell surface biotinylation assay in HEPG2 cells transfected as indicated (Z/R, ZNRF3/RNF43). **n** Co-IF of RSPO2-ZNRF3-FGFR4 in HEPG2 cells. White arrowheads, colocalized FGFR4-RSPO2; white arrow, colocalized FGFR4-RSPO2-ZNRF3; yellow arrow, colocalized FGFR4-RSPO2-ZNRF3 in magnified inset; Dashed lines, nucleus. Scale bar, 20 μm. **o** Model showing (1) RSPO2-LGRs-ZNRF3-FGFR4 complex at the cell surface and (2) membrane clearance of FGFR4 by endocytosis. **p** WISH of *pitx2c* in *Xenopus* tailbuds (St. 29). Arrowheads, *pitx2c* at the LPM. Scale bar, 0.5 mm. **q** Quantification of (**p**). Two-sided Fisher's exact test used for statistical analysis. *n* = number of embryos. Data information: For all Co-IF analyses **h**, **j**, **l** data are displayed as mean ± SD with two-tailed unpaired *t*-test. *n* = number of cells. Source data are provided as a Source Data file.

pERK1/2 levels on both sides (Supplementary Fig. 7e–g) while *rspo2* Mo increased pERK1/2 level on the left side (Supplementary Fig. 7h–j), i.e. LRO FGF signaling became symmetric. Finally, gastrocoel injection of TK peptide, which specifically blocks Rspo2-Fgfr4 interaction, phenocopied *rspo2* Morphants in abolishing LR asymmetric pERK1/2 levels (Supplementary Fig. 7k). Collectively, we establish (i) that an FGF signaling gradient exists across the right- to left side of the LRO, which (ii) depends on Rspo2 acting as FGF signaling inhibitor on the left side (Fig. 7i).

## Discussion

We report that in *Xenopus*, Rspo2 serves as the elusive Hirokawa-signal predicted by the chemosensory model for LR symmetry breakage, promoting the development of sinistral cell fates (Fig. 8a, b). The crucial mechanistic insight is that Rspo2 functions as an antagonist to FGF receptors, leading to the internalization of FGFR4 through ZNRF3-mediated endocytosis. Our conclusions are drawn from several key observations in *Xenopus*: Rspo2 (i) is necessary and sufficient to sinistralize cell fates by counteracting dextralizing FGF signals, (ii) acts downstream of leftward flow but upstream of asymmetric *dand5* expression, and (iii) establishes a sinistro-dextral FGF/ERK signaling gradient in the LRO. Thus, similar to the Spemann-Mangold organizer, where BMP- and Wnt morphogen gradients are generated via secreted antagonists such as Chordin and Dkk1[64,65], Rspo2 is not an instructive signal but instead it restricts the dextralizing function of FGF signaling. In other words, in the absence of an instructive FGF signal, the default cell fate of the embryo is sinistral (Pitx2/Nodal-positive LPM). Rspo2's action is similar to that of Dkk1[66], both act on the respective growth factor receptor instead of on the ligand to create a signaling sink.

How does symmetric Rspo2 expression in the LRO result in asymmetric *dand5* and *pitx2c* expression? The results suggest that leftward flow transports Rspo2 protein to the left side of the LRO, where it removes Fgfr4 from the cell surface and reduces FGF signaling. Simulations predict that the sinistralizing signal, i.e. Rspo2, should be limiting or have a short active lifetime; otherwise a uniform concentration would build up across the LRO, abrogating a symmetry-breaking function[67]. Our observation that only left- but not right-side *rspo2* Mo injection abrogates *pitx2c* expression also suggests that Rspo2 is present in limiting amounts: Following left side Mo injection, ciliary flow apparently fails to transport enough Rspo2 from the right side to the left to compensate for the complete loss of Rspo2 from the very side where it matters. In contrast, FGF can be in excess: since Rspo2 inhibits downstream signaling, it will override the effect of FGF ligand that might also accumulate on the left. Visualizing accumulation of Rspo2 protein will provide more definitive proof for this model in the future. The FGF signaling gradient shows at least 6-fold difference across the LRO, consistent with simulations[23]. Similar morphogen gradient slopes were reported for *Xenopus* antero-posterior β-catenin/Wnt signaling (6-fold)[68] and zebrafish dorso-ventral pSmad5/BMP signaling (5-fold)[69].

Evidence that Rspo2 functions by FGF inhibition are in vitro binding studies, elucidation of the molecular mechanism involving E3 ligase-mediated FGFR4 endocytosis, identification of the FGFR4-TSP1 binding site, and blocking Fgfr4-Rspo2 interaction with TSP1-derived peptides. Of note, while Rspo2 is a poor antagonist of Fgfr1, a role in antagonizing Fgfr2, Fgfr3, or Fgfrl1 and their involvement in flow stage signaling was not tested and cannot be excluded. Notably, Fgfr2 regulates LR specification in zebrafish[70] and *FGFR3* is a heterotaxy gene in humans[71,72].

Indicators that Rspo2 functions in *Xenopus* LR specification as FGF inhibitor are that at LRO-stage, Fgfr4 signaling acts as a dextralizing signal that represses left LPM specification, rescue of LR defects in *rspo2* Morphants by Fgfr4 activation, the sinistro-dextral FGF-ERK1/2 signaling gradient in the LRO, which is abrogated in *rspo2* Morphants, and recapitulation of LR defects with TK peptide, which specifically blocks Rspo2-Fgfr4 interaction. Rspo2 inhibits asymmetric expression of the Nodal antagonist *dand5* and therefore presumably acts by derepressing the Nodal-Pitx2c cascade on the left. In other words, FGF/anti-FGF signaling triggers a downstream anti-Nodal (Dand5)/Nodal vector. Asymmetric expression of *dand5* depends on posttranscriptional regulation by Dicer[16,73]. Following FGF19 gastrocoel injection, *dand5* expression became symmetric, as in *Dicer* knockdown, while Fgfr4 inhibition abolished *dand5* expression (Supplementary Fig. 8a-d). These results suggest a mechanistic link between FGF signaling and posttranscriptional *dand5* mRNA decay, which requires further investigation.

How conserved may FGF/anti-FGF signaling be in other vertebrates? FGF signaling is implicated in LR asymmetry in all vertebrate models analyzed[2,20,33–35,52,53,74–77]. However, consecutive FGF-dependent processes in the LR cascade spanning from mesoderm formation and gastrulation to LRO specification, ciliogenesis, and ultimately symmetry breaking, introduce complexity to the comprehension of its exact function. This complexity has resulted in findings that may appear contradictory because different embryonic stages or regions were interrogated. Adding to this complexity is the multitude of 19 FGF ligands and 5 receptors where different receptor-ligand combinations may trigger different downstream pathways. To address the role of FGF signaling at flow stages, it is therefore essential to exclude earlier effects. We leveraged the opportunities in *Xenopus* of stage-, site-, and LR-specific experimentation, and employing inhibitory peptides to manipulate Rspo2-FGF antagonism. We show that flow-stage Fgfr4 signaling blocks the Nodal-Pitx2 module and that Rspo2 overcomes this inhibition. Our results therefore support the *Release-of-Repression* (RoR) model obtained in rabbit embryo[34,35] (Fig. 8a), whereby FGF at flow stages is a dextralizing signal that blocks the Nodal-Pitx2 cascade bilaterally, while leftward flow unilaterally attenuates this repression on the left side. In chick and rabbit, FGF8-soaked beads promote rightward LPM[33,35], consistent with our data. Hence, FGF as a dextralizing signal at flow stages is evolutionary conserved between amphibia to mammals. Yet, in mouse, FGF8 was proposed as a sinistralizing

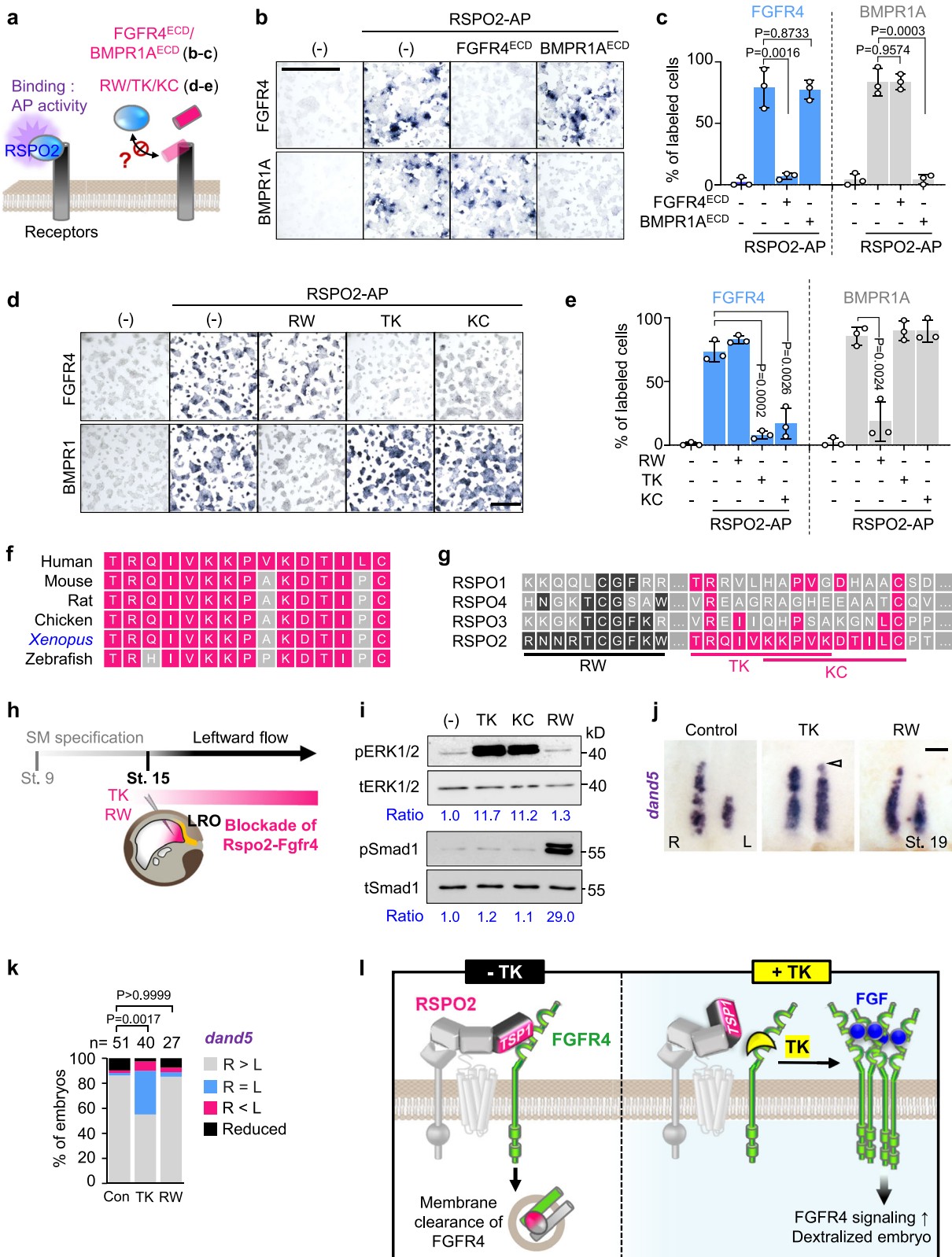

signal[74], apparently contradicting a conserved role of FGF as dextralizing signal. However, the genetic evidence rests on constitutive *Fgf8* mutants that display axial abnormalities, and hence indirect effects due to early requirement for FGF signaling may not be excluded. In mouse *Rspo2*[78] and *Fgfr4*[79] mutants, LR defects have not been reported but laterality defects could have been missed, because they may not affect viability. We note though, that human

*RSPO2* deficiency causes congenital heart defects[78], a condition associated with LR misregulation[4].

Recent elegant micromanipulation experiments support that leftward flow mechanically activates PKD-Ca$^{2+}$ signaling by bending non-motile cilia on the left side for LR specification, strongly supporting a mechanical read-out of leftward flow[27,28]. However, chemosensation and mechanosensation in LR specification are not mutually

**Fig. 6 | TK peptide disrupts RSPO2-FGFR4 interaction and derepresses *dand5* in the LRO. a** Scheme of cell surface competitive binding assays in (**b**–**e**). **b** Representative images of HEK293T cells transfected and treated as indicated. Scale bar, 1 mm. **c** Quantification of (**b**). **d** Representative images of HEK293T cells transfected and treated as indicated. Scale bar, 1 mm. **e** Quantification of (**d**). **f** Amino acid sequence comparison of TK-KC peptides in RSPO2 TSP1 domain of several species. Note that TK-KC peptide sequence is highly conserved (Magenta boxes). **g** Amino acid sequence comparison of RSPO1-4 TSP1 domains. Note that TK-KC peptide sequence derived from human RSPO2 TSP1 domain is unconserved in other RSPOs (Magenta boxes). **h** Microinjection strategy for (**i**–**k**). **i** Western blot

analysis of phosphorylated ERK1/2 and Smad1 (pERK1/2 and pSmad1) and total ERK1/2 and Smad1 (tERK1/2 and tSmad1) in *Xenopus* embryo (St. 15) lysates. **j** WISH of *dand5* in *Xenopus* LRO (St. 19) injected as indicated. Arrowheads, *dand5* derepression. Scale bar, 100 µm. **k** Quantification of (**j**). Two-sided Fisher's exact test used for statistical analysis. *n* = number of dorso-posterior explants. **l** Model showing the mode of action for TK to intervene RSPO2-FGFR4 interaction and increases FGFR4 signaling. Data information: For all cell surface binding assays (**c**, **e**), data are displayed as mean ± SD with two-tailed unpaired *t*-test: *n* = 3 biologically independent samples. Source data are provided as a Source Data file.

exclusive and Rspo2 may act in concert with Ca²⁺ signaling, which will be interesting to test in the future.

Looking beyond LR specification, our study reveals a surprising trifunctionality of RSPO2, acting as agonist of Wnt- and antagonist of BMP and FGF signaling. Thus, RSPO2 functions as a multimodal logical switch of extracellular signals: Wnt (ON), BMP (OFF), FGF (OFF). Depending on where *RSPO2* and its receptor targets are expressed, RSPO2 may regulate all three signals simultaneously, such as in the *Xenopus* LRO, a region where posteriorizing Wnt, ventralizing BMP, and lateralizing FGF coincide[32,51,52,68,80]. The multimodal function may also pan out in cancer, notably colorectal (CRC) where RSPO2 is thought to act merely as Wnt agonist but where FGF and BMP signaling are also implicated[81–84]. The availability of inhibitory peptides that selectively interfere with RSPO2's BMPR1A or FGFR4 antagonism, will aid such future analysis.

Our study is limited to *Xenopus*; further verification is required to establish the relevance of Rspo2 and the FGF/anti-FGF signaling axis in LR specification in other species. Direct visualization is needed to observe the predicted leftward accumulation of Rspo2 protein in the LRO. Finally, the identity of the dextralizing FGF ligands and -receptors other than Fgfr4, remains unknown.

## Methods

### *Xenopus laevis*
All *Xenopus laevis* experiments were approved by the state review board of Baden-Württemberg, Germany (permit number 35-9185.81/G-141/18 and G-116/23 (Regierungspräsidium Karlsruhe)) and performed according to the federal and institutional regulations. Adult wild-type *Xenopus laevis* frogs were purchased from Nasco, the National *Xenopus* Resource (NXR), and the European Xenopus Resource Centre (EXRC). Developmental stages of *Xenopus* embryos were determined according to Nieuwkoop and Faber (https://www.xenbase.org). In vitro fertilization and culture of *Xenopus laevis* embryos were performed according to the standard protocol (https://www.xenbase.org).

### Human cell lines
HEPG2 and HEK293T cells (ATCC HB-8065 and ATCC CRL-3216) were cultured in DMEM High glucose (Gibco 11960 or Capricorn DMEM-HXA) supplemented with 10% FBS (Capricorn FBS-12A9), 2 mM L-glutamine (Sigma G7513) and 1% penicillin-streptomycin (Sigma P0781). H1581 cells (gift from Dr. R.Thomas) were cultured in RPMI (Gibco 21875 or Capricorn RPMI-XA) supplemented with 10% FBS, 2 mM L-glutamine, 1 mM sodium pyruvate (Sigma S8636) and 1% penicillin-streptomycin. Regular mycoplasma test guaranteed all cell lines were mycoplasma negative.

### *Xenopus laevis* microinjection
For mRNA and antisense morpholino oligonucleotide (Mo) injection, *Xenopus laevis* 2–4 cell stage embryos were microinjected 5 nl per each embryonic blastomere with in vitro transcribed mRNAs or Mo using the Harvard Apparatus microinjection system. pCS2+ plasmids inserted with Rspo2$^{\Delta C}$-myc, Rspo2$^{\Delta FU1}$-myc, Rspo2$^{F107E}$-myc, Rspo2$^{\Delta TSP1}$-myc, Bmpr1a$^{DN}$, Fgfr4-EYFP, human ZNRF3, and Znrf3$^{\Delta RING}$ were used as templates for in vitro transcribed mRNAs. In vitro transcription was

performed as previously described[31]. Mos targeting *rspo2*[32], *lrp6*[32], *znrf3*[32], *gas2l2*[41] and standard control[32] were purchased from Gene-Tools. Equal amounts of total mRNA or Mo were injected by adjustment with *preprolactin* (*ppl*) mRNA or standard control Mo. Injected stages and position of the embryos are indicated in the figure legends. Randomization of microinjection order was not applied during the experiments. Statistical analysis was not adjusted to sample size before microinjection. For amounts of injected mRNAs and Mos, see Injected amount of reagents per *Xenopus* embryo.

### *Xenopous laevis* gastrocoel microinjection
*Xenopus laevis* St. 15 embryos were microinjected 10-20 nl per each gastrocoel cavity with 2 ng of human RSPO2 (R&D systems 3266), 1 ng of Wnt3A (Elabscience PKSH033972) and 2 ng of DKK1 (Peprotech 120-30) recombinant proteins dissolved in 1x MR (Modified Ringer's) solution or 50 µM TK, KC and RW monomeric peptides[63] dissolved in 0.1x MR solution. DMSO was injected as a control for 2 µM BLU9931 (Calbiochem 5387760001). To mediate leftward flow defects, *Xenopus* St. 15 embryos were microinjected 20 nl per each gastrocoel cavity with 1.5% methylcellulose (Sigma M0430) dissolved in 0.1x MR. Randomization of microinjection order was not applied during the experiments. Statistical analysis was not adjusted to sample size before microinjection.

### *Xenopus laevis* organ laterality analysis
To monitor heart looping, microinjected St. 42 embryos were transferred to 0.02 % Tricaine/MS222 (Sigma 886862) for anesthesia and scored for quantification under the light microscope. Heart looping defect was determined by the position of outflow tracts. Representative hearts were dissected after fixation with MEMFA for 30 min. For investigating gut looping, microinjected St. 42 embryos were fixed with MEMFA for 2 h and guts were dissected for analysis. Gut looping defects were determined by the position of midgut loop concavities. Dissected hearts and guts were placed on fresh 1% agarose gels from ventral view and images were obtained using AxioCam MRc 5 microscope (Zeiss) and processed with AxioVision 40 version 4.8.2.0 software. Background of images was adjusted using Removing background tool of Microsoft PowerPoint 2019 software and pasted into the uniform background color for presentation.

### *Xenopus laevis* whole-mount in situ hybridization
In situ hybridization of *Xenopus laevis* whole embryos and dissected dorso-posterior explants were performed utilizing digoxigenin (DIG)-labeled antisense probes according to the standard protocol (http://www.xenbase.org). In brief, embryos and the LRO explants were fixed in MEMFA for 2 h and dehydrated in MeOH, followed by rehydration and hybridization with 250ng-1µg of DIG-labeled probes overnight at 65 °C. Embryos and explants were treated with 1:4000 diluted anti-DIG antibody (Sheep anti-digoxigenin-AP, Roche 11093274910) for 4 h at room temperature. After thorough washing, gene expression was visualized by BM purple AP substrate (Sigma 11442074001). DIG-labeled probes against *rspo2*, *rspo3*, *sizzled*, and *znrf3* were generated as previously described[31,32]. Probes against *pitx2c, nodal* and *dand5* were prepared using pBSKII-*Xenopus laevis* Pitx2c plasmid (Gift from

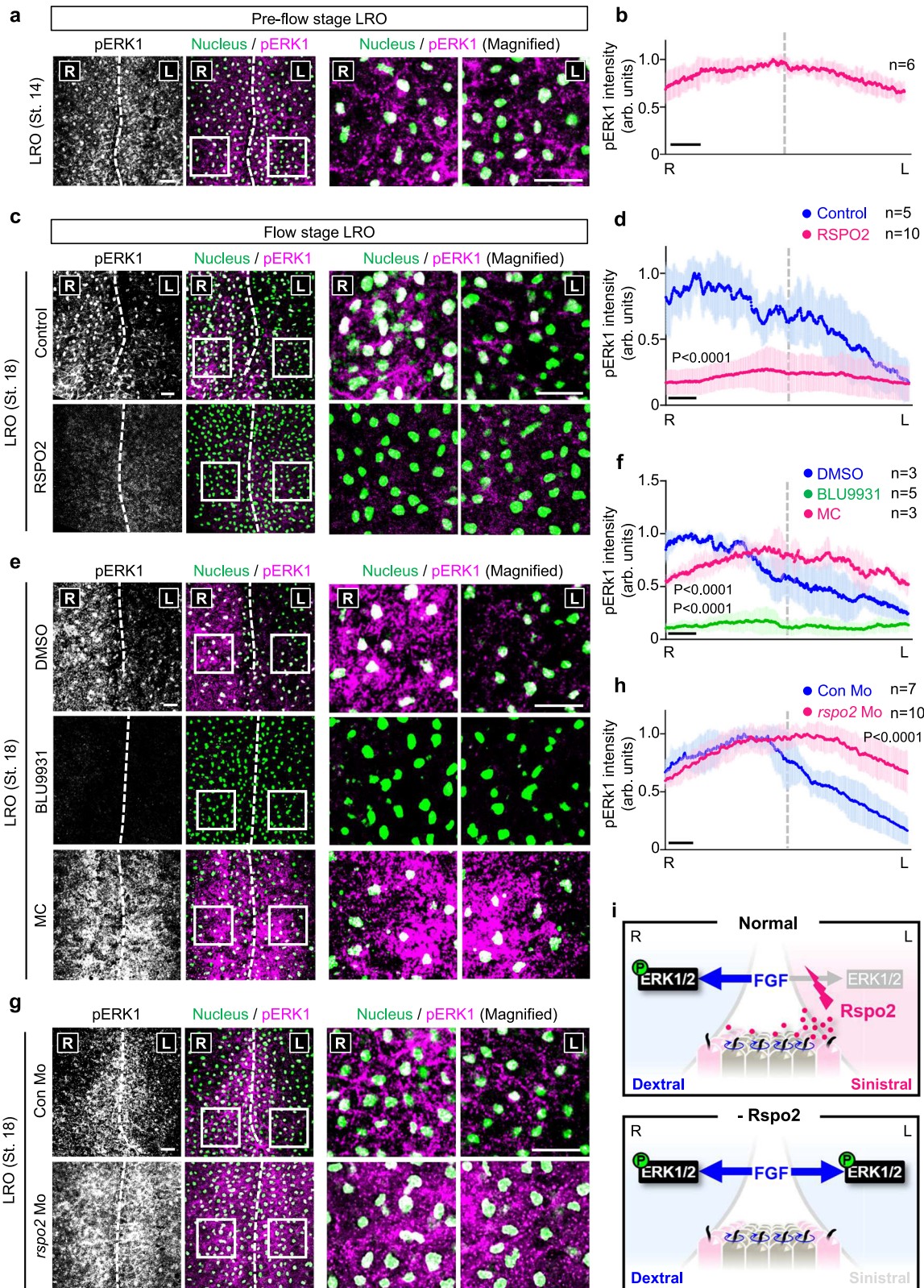

**Fig. 7 | A LR-FGF signaling gradient in the LRO that requires Rspo2 and ciliary flow. a, c, e, g** Immunofluorescence microscopy (IF) for phosphorylated ERK1 (pERK1) in the LRO from *Xenopus* St.14 (**a**) or St. 18 (**c**, **e**, **g**) neurula. Nuclei were stained with Hoechst. R right, L left. Dashed line, midline in the LRO. White boxes are magnified in right panels. Scale bar, 50 µm. **b, d, f, h** Normalized pERK1 intensity profile for the LRO explants in (**a**, **c**, **e**, **g**). *n* = number of LRO explants. Blue, magenta, and green lines, mean pERK1 intensity. Blue, magenta, and green shades, ±SD. The highest mean pERK1 intensity on the right side of control LRO (**b**, **d**), DMSO-injected LRO (**f**), or control Mo-injected LRO (**h**) was set to 1. R right, L left. Dashed line, dorsal midline. Scale bar, 50 µm. **i** Anti-FGF signaling by Rspo2 is required to establish a LR-FGF signaling gradient in the LRO. Data information: Two-tailed paired *t*-test was used for all statistical analyses. Source data are provided as a Source Data file.

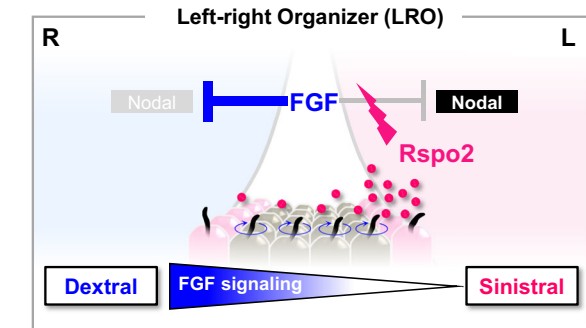

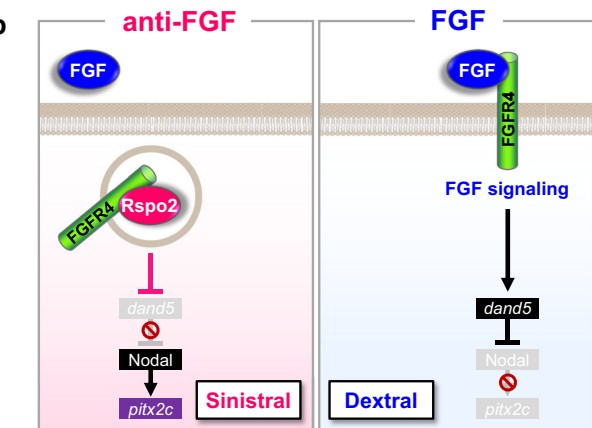

**Fig. 8 | Rspo2 regulates LR symmetry breakage by establishing an FGF signaling gradient. a** LR symmetry breakage in the flow-stage LRO. FGF signaling blocks the Nodal-Pitx2 cascade. Leftward flow presumably transports secreted Rspo2 protein to the left side where it accumulates and inhibits FGF signaling, generating a right-to-left FGF signaling gradient to release the Nodal-Pitx2 cascade. **b** Molecular cascade of LR symmetry breakage in the flow-stage LRO. Rspo2 protein internalizes Fgfr4 on the left side, thereby desensitizing cells to FGF signaling and derepressing the Nodal-Pitx2 cascade. Fgfr4 removal attenuates FGF signaling even if FGF ligand also accumulates on the left.

Dr. Axel Schweickert), Nodal plasmid and Dand5 plasmid (Horizon discovery 6324534) as templates. Probes against *rspo1, fgfr1, fgfr4* and *lgr4* were generated using pBSKII-*Xenopus tropicalis* Rspo1, pCS2 + -*Xenopus laevis* Fgfr1, pCS2 + -*Xenopus laevis* Fgfr4, and pCMV-Sport-*Xenopus tropicalis* Lgr4 plasmids as templates, respectively. Probe against *fgf19* was prepared using *Xenopus tropicalis* Fgf19 plasmid (Gift from Dr. Karel Dorey). *dand5* at the LRO was categorized by asymmetric expression patterns. R > L showed normal *dand5* repression in the left LRO margin, R = L showed *dand5* derepression in the left LRO margin, R < L showed reversed *dand5* expression and Reduced, decreased *dand5* expression. Representative images were obtained using AxioCam MRc 5 microscope (Zeiss) and processed with AxioVision 40 version 4.8.2.0 software. Embryos in each image were either selected using Magic Wand tool of Adobe Photoshop CS6 software version 13.0, or selected using Removing background tool of Microsoft PowerPoint Standard 2019 software and pasted into the uniform background color for presentation. Phenotypic scoring of embryos was unblinded and executed twice.

### Induction of iFgfr in *Xenopus laevis*

iXFGFR4-pCS2+ (Addgene 31258) and iXFGFR1-pCS2+ (Addgene 31257)[55] were linearized with NotI restriction enzyme and in vitro transcribed with SP6 RNA polymerase (Invitrogen AM1340) to produce *ifgfr4* and *ifgfr1* mRNAs for microinjection. For iFgfr4 induction during leftward flow stages, 20 nl of 1 µM AP20187 (Sigma SML2838) was injected into the gastrocoel cavity of St. 15 embryos. For rescue of *rspo2* derived LR asymmetry defects, embryos co-injected with *rspo2* and *ifgfr* mRNAs were transferred to 0.1x MR (Modified Ringer's) solution supplemented with 1 µM AP20187 at St. 10. Embryos were rinsed carefully and transferred to 0.1x MR solution without AP20187 at St. 12-13 and harvested for in situ hybridization.

### *Xenopus laevis* western blot analysis

Microinjected *Xenopus laevis* embryos, animal cap explants and LRO explants were harvested at indicated stages and homogenized in NOP+ lysis buffer (20 mM Tris-HCl pH 7.5, 150 mM NaCl, 2% Triton X-100, 0.2% DTT, and cOmplete Protease Inhibitor Cocktail (Roche 11697498001)) with a volume of 20 µl per embryo, 5 µl per animal cap explant, or 4 µl per LRO explant. Lysates were cleared by adding CFC-113 (Honeywell 34874), followed by centrifugation and boiling at 95 °C for 5 min with NuPAGE LDS Sample Buffer (ThermoFisher NP0008) containing 50 mM DTT. 1:1000 dilution of primary antibodies in 5% BSA-TBST (TBS with 0.1% Tween-20) were applied to transferred membranes and incubated overnight at 4 °C. 1:10000 dilution of HRP-linked secondary antibodies were applied for 1 h at room temperature and images were obtained with SuperSignal West pico ECL (Thermo-Fisher 34580) using LAS 3000 version 2.21 system (FujiFilm). Quantification of immunoblots was performed using FIJI (Image J) v1.51k software. Primary antibodies used for *Xenopus laevis* western blotting: Rabbit anti-Phospho-ERK1/2 (Cell Signaling Technology 9101 S); Rabbit anti-ERK1/2 (Sigma M5670); Rabbit anti-Phospho-Smad1 (Cell Signaling Technology 9516); Rabbit anti-Smad1(Cell Signaling Technology 9743 S); Rabbit anti-GAPDH (Cell Signaling Technology 2118 S); Mouse anti-active-beta-catenin (Millipore 05-665); Rabbit anti-ERK1/2 (Gene-Tex GTX134462). Secondary antibodies used for *Xenopus laevis* western blotting: Goat anti-mouse IgG (H + L) HRP (Jackson ImmunoResearch 115-035-146); Goat anti-rabbit IgG (H + L) HRP (Jackson ImmunoResearch 111-035-144). Detailed information on antibodies used is available in Reporting Summary.

### *Xenopus laevis* immunofluorescence

For phosphorylated ERK1 (pERK1) staining, microinjected *Xenopus laevis* LRO explants were bisected at St. 14 or St. 18 and fixed immediately in MEMFA for 2 h at room temperature. After three times of washing with PBS, explants were permeabilized with 0.5% Triton X-100 in PBS overnight at 4 °C. Explants were incubated with blocking solution (10% normal donkey serum, 5% BSA, 0.2% Tween 20 in PBS) for 2 h at room temperature, followed by 1:250 dilution of mouse anti-Phospho-ERK1 (Santa Cruz sc-7383) antibody treatment overnight at 4 °C. 1:500 dilution of goat anti-mouse Alexa Fluor 546 (Invitrogen A11030) and Hoechst dye were further applied for 2 h at room temperature. After extensive washing with PBS, explants were mounted with Fluoromount-G (ThermoFisher 00495802) and images were obtained using LSM 700 (Zeiss) confocal microscope and processed with Zeiss ZEN 2012 (black edition) version 2.5.

For pERK1 quantification, fluorescence intensity profiles were analyzed from raw images of pERK1 immunofluorescence using the plot profile command in FIJI (Image J) v1.51k software. The pixel intensity value corresponding to pERK1 was measured across the right to the left axis of a 402 µm × 402 µm square (644 datapoints, St. 14) or a 424 µm × 424 µm square (680 datapoints, St. 14) for each LRO explant. Mean pixel intensities were plotted in the figures with standard deviations (SD) indicated. The highest pERK1 intensity on the right side of control or control Mo injected embryos was set to 1. The position of the LRO midline was determined from brightfield images obtained using LEICA DMIL microscope/Canon DS126311 camera.

For monitoring cell surface Fgfr4 level in *Xenopus laevis*, *fgfr4*-EYFP and membrane-RFP mRNAs were co-injected with the indicated mRNAs animally in the 4-cell stage embryos. Animal cap explants were dissected from injected embryos at St. 9 and immediately fixed with 4% Paraformaldehyde for 2 h and mounted using LSM 700 (Zeiss). Images

were processed with Zeiss ZEN 2012 (black edition) version 2.5. For quantification, Pearson's correlation coefficient for EYFP and RFP was analyzed using FIJI (Image J) v1.51k software. Six to eighteen random areas harboring 10–15 cells chosen from 2–8 embryos were quantified per each set. Detailed information on antibodies used is available in Reporting Summary.

## Injected amount of reagents per *Xenopus* embryo

Equal amounts of total RNA or Mo were injected by adjustment with *preprolactin* (PPL) mRNA or standard control Mo. Per embryo; Fig. 1f and i, 50 and 100 pg of *rspo2* mRNA; Fig. 1g and j, 10 ng of *rspo2* Mo; Fig. 1m and p, 8 ng of *rspo2* Mo; Fig. 2b and d, 100 and 150 pg of *rspo2* mRNA; Fig. 2f and h, 100 and 150 pg of *rspo2* mRNA and 5 ng of *gas2l2* Mo; Fig. 3i, 200 pg of *ifgfr4* mRNA; Fig. 3k, 250 ng of *rspo2* and 200 pg of *ifgfr1/4* mRNA; Fig. 3l, 10 ng of *rspo2* Mo; Fig. 4d, 10 ng of *rspo2* Mo, 5 ng of *lrp6* Mo and 200 pg of *bmpr1a*$^{DN}$ mRNA; Fig. 5q, 80 ng of *znrf3* Mo and 250 pg of *znrf3* mRNA; Fig. 7g, 10 ng of *rspo2* Mo; Supplementary Figure 1f, 25 pg of *rspo2* and *rspo2* mutants mRNA; Supplementary Figure 1i and 1j, 200 pg of *bmpr1a*$^{DN}$ mRNA; Supplementary Figure 1k, 10 ng of *rspo2* Mo and 200 pg of *bmpr1a*$^{DN}$ mRNA; Supplementary Figure 1l and 1n, 200 pg of *ifgfr1/4* mRNA; Supplementary Figure 2h, 50 and 100 pg of *rspo2* mRNA; Supplementary Figure 5g, 200 pg of *fgfr4*-EYFP, 100 pg of membrane-RFP mRNA, 250 pg of *rspo2* and mutant *rspo2* mRNA; Supplementary Figure 5j, 200 pg of *fgfr4*-EYFP and 100 pg of membrane-RFP mRNA, 250 pg of *rspo2* and *znrf3*$^{DN}$ mRNA; Supplementary Fig. 6d, 20 pg of TOPFlash DNA, 10 ng of Renilla DNA and 5 ng of *lrp6* Mo.

## Cell transfection

For HEPG2 and H1581 cells, DNA plasmids and siRNAs were transfected by Lipofectamine 3000 (Invitrogen L3000) and DharmaFECT 1 transfection reagent (Dharmacon T-2001) respectively, following the manufacturer's instructions. For HEK293T cells, DNA plasmids were transfected using X-tremeGENE 9 DNA transfection reagent (Roche 6365787001) according to the manufacturer's instructions. siRNAs used for transfection: siControl (Dharmacon D-001210-01-20), si*RSPO2* (Dharmacon M-017888-01), si*LRP5* (Dharmacon M-003844-02), si*LRP6* (Dharmacon M-003845-03), si*ZNRF3* (Dharmacon M-010747-02), si*RNF43* (Dharmacon M-007004-02), si*LGR4* (Dharmacon M-003673-03), si*LGR5* (Dharmacon M-005577-01), si *ßCatenin* (Dharmacon M-003482-00). si*BMPR1A* (Dharmacon M-004933-04).

## Production of conditioned medium

HEK293T cells were seeded in 15 cm culture dishes and transfected with 2 µg of human RSPO1-4-AP, RSPO2 mutants-AP, RSPO1-4-flag or *Xenopus* Rspo2 wildtype and mutants-AP plasmids. After 24 h, media were changed with fresh DMEM, 10% FBS, 1% L-glutamine and 1% penicillin-streptomycin and cultured 4 days at 32 °C. Conditioned media were harvested two times every 2 days. Produced media were validated with western blot analyses, TOPFlash assays or AP activity measurement. Wnt3A conditioned medium was produced from mouse L-cells stably transfected with Wnt3A (ATCC CRL-2647) and validated with TOPFlash assays.

## Western blot analysis

For HEPG2 cells, cultured cells were serum starved for 24 h and stimulated 30 min with 10 ng ml$^{-1}$ recombinant human FGF19, FGF21, or FGF23 protein (Abcam ab50132, Peprotech 100-42, Peprotech 100-52) or RSPO1-4 conditioned media. In Supplementary Figure 2m and n, cells were treated with 300 nM LDN193189 (Tocris 6053) and 250 ng ml$^{-1}$ recombinant murine Noggin protein (Peptrotech 250-38). For H1581 cells, cultured cells were transfected with 50 nM of indicated siRNAs for 48 h and stimulated with 2 ng ml$^{-1}$ recombinant human FGF19 protein (Abcam ab50132). In Fig. 4g, 500 ng ml$^{-1}$ BMPR1A-HA plasmid and GFP (MOCK) plasmid were transfected. In Fig. 4i,

H1581 cells were treated with 2 µM BLU9931 (Calbiochem 5387760001) in DMSO. HEPG2 and H1581 cells were lysed in Triton lysis buffer[32] supplemented with cOmplete Protease Inhibitor Cocktail (Roche 11697498001) and PhosSTOP (Roche 4906845001). 10-20 µg of lysates were mixed with NuPAGE LDS Sample Buffer (ThermoFisher NP0008) containing 50 mM DTT and boiled at 95 °C for 5 min. 1:1000 dilution of primary antibodies in 5% BSA-TBST (TBS with 0.1% Tween-20) were applied to transferred membranes and incubated overnight at 4 °C. 1:10000 dilution of HRP-linked secondary antibodies were applied for 1 h at room temperature and images were obtained with SuperSignal West pico ECL (ThermoFisher 34580) using LAS 3000 version 2.21 system (FujiFilm). Quantification of immunoblots was performed using FIJI (Image J) v1.51k software. Primary antibodies used for western blotting: Rabbit anti-Phospho-ERK1/2 (Cell Signaling Technology 9101 S); Rabbit anti-ERK1/2 (Sigma M5670); Mouse anti-beta-Catenin (BD 610154); Rabbit anti-LRP6 (Cell Signaling Technology 2560); Rabbit anti-GAPDH (Cell Signaling Technology 2118 S); Goat anti-RSPO2 (R and D systems AF3266); Rat anti-HA (Roche 11867423001); Rabbit anti-Phospho-Smad1 (Cell Signaling Technology 9516); Rabbit anti-Smad1(Cell Signaling Technology 9743 S). Secondary antibodies used for western blotting: Goat anti-mouse IgG (H + L) HRP (Jackson ImmunoResearch 115-035-146); Goat anti-rabbit IgG (H + L) HRP (Jackson ImmunoResearch 111-035-144); Mouse anti-goat IgG (H + L) HRP (Jackson ImmunoResearch 205-035-108). Detailed information on antibodies used is available in Reporting Summary.

## Luciferase reporter assays

BRE luciferase assays and Gal luciferase assays were executed as previously described. In brief, HEPG2 cells were transfected with reporter plasmids and serum starved overnight. Cells were stimulated 20 h with 100 ng ml$^{-1}$ recombinant human FGF19 protein (Abcam ab50132) along with conditioned medium of RSPO2 wild-type or RSPO2 mutants. TOPFlash assay was executed as previously described using HEK293T cells[32] or *Xenopus laevis* embryos[63]. Luciferase activity was measured with the Dual luciferase reporter assay system (Promega E1960) and Thermo Fluoroskan Ascent Software version 2.6. Firefly luminescence was normalized to Renilla.

## Cell surface binding assay

HEK293T cells were seeded in 24 well plates coated with Poly-D-Lysine (Sigma P6407). 250 ng ml$^{-1}$ of human FGFR4-HA (Sino Biological HG10538-CY), BMPR1A-HA, *Xenopus laevis* Fgfr4a-V5, and *Xenopus tropicalis* Lgr4-V5 DNA were transfected in HEK293T cells and incubated with 2-2.5 U ml$^{-1}$ AP-fused conditioned media for 3 h on ice. For competitive binding assays, cells were treated with RSPO2-AP in combination with either 5 nM human FGFR4 Fc Chimera (R&D systems 685-FR) or 5 nM BMPR1A Fc Chimera (Abcam ab238293). For peptide competition assays, cells were treated with RSPO2-AP in combination with 50 µM TK, KC and RW peptides. Receptors-RSPO binding was crosslinked with dithiobis (succinimidyl) propionate (DSP) (Thermo 22585) for 15 min on ice and additional 30 min at room temperature. Cells were washed three to four times with DPBS and treated with 2 mM Levamisole for 15 min to inactivate endogenous AP activities and developed with BM-Purple (Sigma 11442074001). Cells were mounted with Fluoromount G (Invitrogen 00495802). Representative images were obtained using LEICA DMIL microscope/Canon DS126311 camera.

## In vitro binding assay

High binding 96-well plates (Greiner M5811) were coated with 2–4 µg ml$^{-1}$ of recombinant human FGFR4 Fc Chimera (R&D systems 685-FR and BioLegend 752502), FGFR1 Fc Chimera (Peprotech 160-02), or RSPO2 (Peprotech 120-43) reconstituted in bicarbonate coating buffer (50 mM NaHCO$_3$, pH 9.6) overnight at 4 °C. Coated wells were washed several times with TBST (TBS with 0.1% Tween-20) and blocked with 5% BSA in TBST for 1 h. 2 U ml$^{-1}$ of RSPOs-AP was incubated

overnight at 4 °C. For in vitro competitive binding assay, coated wells were treated with 2 U ml$^{-1}$ RSPO2-AP in combination with 100 μM of monomeric peptides for 3 h at room temperature. Wells were washed several times with TBST and bound AP activity was measured by AquaSpark AP substrate (Serva 42593.01) using Thermo Fluoroskan Ascent Software version 2.6. $K_d$ was obtained with In vitro binding assay using recombinant human FGFR4 Fc Chimera (R&D systems 685-FR) and RSPO2-AP[32,60].

### Cell surface biotinylation assay
Cell surface biotinylation of HEPG2 cells was executed as previously described[32]. In brief, HEPG2 cells were transfected with 50 nM of indicated siRNAs for 48 h and treated with RSPO2 for 30 min. H1581 cells were transfected with 50 nM of indicated siRNAs for 72 h. Surface proteins were biotinylated with 0.25 mg ml$^{-1}$ sulfo-NHS-LC-LC-Biotin (ThermoFisher 21338). 250 μg of lysate was incubated with 20 μl streptavidin agarose (ThermoFisher 20359) to pull-down biotinylated proteins. Biotinylated proteins were subjected to SDS-PAGE applying with 1:1000 dilution of rabbit anti-FGFR4 (Cell Signaling Technology 8562 S) or rabbit anti-Transferin receptor (Cell Signaling Technology 13113 S) and 1:10000 dilution of goat anti-rabbit IgG (H + L) HRP (Jackson ImmunoResearch 111-035-144). Quantification was performed using FIJI (Image J) v1.51k software. Detailed information on antibodies used is available in Reporting Summary.

### Surface receptor internalization assay
Surface receptor internalization was monitored in HEPG2 cells as previously described[32]. In brief, surface proteins were biotinylated with 0.5 mg ml$^{-1}$ sulfo-NHS-SS-Biotin (ThermoFisher 21331). After quenching, pre-warmed control medium or RSPO2 conditioned medium was added at 37 °C for 30 min. Remaining surface-biotin was removed by 50 mM MesNa (2-mercaptoethanesulfonate, CAYMAN 21238). 500 μg of lysate was incubated with 25 μl streptavidin agarose (ThermoFisher 20359) to pull-down biotinylated proteins (internalized proteins) and subjected to SDS-PAGE applying with 1:1000 dilution of rabbit anti-FGFR4 (Cell Signaling Technology 8562 S) or rabbit anti-Transferin receptor (Cell Signaling Technology 13113 S) and 1:10000 dilution of goat anti-rabbit IgG (H + L) HRP (Jackson ImmunoResearch 111-035-144). Quantification of blots was executed using FIJI (Image J) v1.51k software.

### Immunofluorescence
For immunofluorescence (IF) of HEPG2 cells, 100,000 cells were grown on coverslips in 12-well plates and incubated with RSPO2-flag conditioned media for 1.5 h. For H1581 cells, 150,000 cells were grown on coverslips in 12-well plates and transfected with 50 nM siRNA for 48 h. HEPG2 and H1581 cells were fixed in 4% PFA for 15 min, blocked and permeabilized with 5% BSA in PBSTB for 1 h and treated with 1:250 diluted rabbit anti-FGFR4 (Cell Signaling Technology 8562 S), mouse anti-EEA1 (BD 610457), mouse anti-clathrin (BD 610499), rabbit anti-FGFR1 (Cell Signaling Technology 9740 S), mouse anti-Flag (Sigma F3156) and mouse anti-Lamp1 (Cell Signaling Technology 15665 S) overnight at 4 °C. 1:500 diluted donkey anti-mouse Alexa Fluor 647 (Invitrogen A31571), donkey anti-rat Alexa Fluor 488 (Invitrogen A21208), donkey anti-rabbit Alexa Fluor 546 (Invitrogen A10036) and goat anti-mouse Alexa Fluor 488 (Invitrogen A11029) and Hoechst dye (1:500) were applied for 2 h at room temperature. Quantification of IF was performed using FIJI (Image J) v1.51k software. Tyramide Signal Amplification to detect RSPO2-HRP was executed as previously described[32]. Representative images were obtained using LSM 700 (Zeiss) and processed with Zeiss ZEN 2012 (black edition) version 2.5.

### Statistics and reproducibility
Sample sizes are reported in each figure and legend. Statistical significance (*P*-value) of analyses was conducted using Graphpad Prism7 software version 7.03. Means between two experimental groups were compared by two-tailed unpaired or paired Student's *t*-test. Results are displayed as mean ± standard deviation (SD). *Xenopus* phenotypes were compared with two-tailed Fisher's exact test. In situ hybridization in Fig. 1b was repeated 3 times independently with similar expression patterns. Cell surface binding assay in Fig. 5b and e was repeated 3 times independently with similar results. Receptor internalization assay in Fig. 5f was repeated 2 times independently with similar results. Western blot analysis in Fig. 5m was repeated 3 times independently with similar results. Confocal microscopy in Fig. 5n was repeated 2 times independently with similar colocalization results. Western blot analysis in Fig. 6i was repeated 3 times independently with similar results.

### Reporting summary
Further information on research design is available in the Nature Portfolio Reporting Summary linked to this article.

## Data availability
Xenbase website (https://www.xenbase.org. RRID:SCR_003280) was used to obtain information about morpholinos and expression patterns. No third-party datasets were analyzed in this study. The authors declare that all data supporting the findings of this study are available within the article and its Supplementary Information files. Further information on research design is available in Reporting Summary. Source data are provided with this paper.

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

## Acknowledgements

We thank A. Schweickert for providing the *Pitx2c* construct; K. Dorey for providing the Fgfr and Fgf constructs; P. Vick for the constructive discussion of the manuscript. We thank C. Seidl for initial cell surface binding studies. We thank A. Hirth and C. Seidl for *Xenopus* purchase and technical support. We acknowledge NXR (RRID: SCR_013731), Xenbase (RRID: SCR_004337) and EXRC (RRID: SCR_007164) for *Xenopus* resources. Technical support by the DKFZ core facility for light microscopy and the central animal laboratory of DKFZ is gratefully acknowledged. This study was funded by Deutsche Forschungsgemeinschaft (DFG) SFB 1324-B01 (C.N.).

## Author contributions

H.L. and C.N. conceptualized the study. H.L conceived, performed, and analyzed the majority of experiments. C.M.C. conducted parts of the Xenopus laevis, human cell line and in vitro experiments. H.L. generated illustrations for schematics and models used in the study. C.N. acquired funding and supervised the project. H.L. and C.N. wrote the manuscript.

## Funding

## Competing interests

The authors declare no competing interests.
