## [Peer Review File · Nature Communications]

R-Spondin 2 governs *Xenopus* left-right body axis formation by establishing an FGF signaling gradientReviewers' Comments:

Reviewer #1:

Remarks to the Author:

Lee et al found that Rspo2 is expressed bilaterally at the Xenopus LRO, and examined its role in L-R asymmetry by Mo injection. Injection of Rspo2 Mo into the left side of the embryos showed more profound effects than the injection to the right side, suggesting that Rspo2 functions mainly on the left-side of the LRO. Since bilateral activation of FGF4 signaling with ifgfrb and AP20187 induced bilateral Dnd5 expression, and this effect was partially suppressed by excess of Rspo2, it was proposed that Rspo2 inhibits FGF4 signaling. Molecular mechanism of Rspo2 action was examined with cultured cells. Those data obtained with cultured cells suggested that i) Rspo2 interacts with FGFR4 protein and inhibits FGF signaling, 2) Rspo2 stimulates the internalization of FGFR4, antagonizing FGF signaling. By combining the data obtained with Xenopus embryos and those with cultured cells, the authors proposed that Rspo2 generates asymmetry in FGF signaling at the LRO.

The paper contains several new interesting findings, such as a new mechanism of Rspo action. However, some of the key data obtained with cultured cells must be tested with embryos whenever feasible, if the authors wish to propose a new LR mechanism in embryos.

- 1) Most importantly, the level of FGF signaling at the LRO of frog embryos must be examined, such as with a pERK antibody, to see if FGF signaling is indeed asymmetric at the LRO in non-treated (wild-type) Xenopus embryos. If FGF signaling is asymmetric, is its asymmetry dependent on Rspo2? This would be the most essential data that is required for the model presented in this paper.
- 2) Internalization of FGFR4 by Rspo2 (Fig. 5g-l), which is another important mechanism proposed by this paper, can be examined with frog embryos. Also, is the level of the extra-cellular domain of FGFR4 on the cell surface reduced by Rspo2? This can be investigated with an antibody against the extra-cellular domain.
- 3) The authors propose that Rspo2 protein is transported toward the left side of LRO by the flow (Fig. 3m, and Discussion). If Rspo2 protein is transported from the right to the left side, why does rspo2 show more profound effects on the left-side?
- 4) Regulation of Dand5 mRNA at the LRO of Xenopus embryo is post-transcriptional (Maerker et al 2021). Does FGF signaling stabilize Dand5 mRNA, in cultured cells or in embryos?
- 5) It has been shown by others (Schneider et al, 2019) that inhibition of FGF signaling by Sprouty impairs specification of the LRO cells (somatic GRP cells), down-regulating the level of Dand5 (and Nodal) mRNA at the LRO of Xenopus embryo. This discrepancy needs to be clarified.
- 6) Whether Rspo2 protein is indeed transported by the flow is the next issue, because it is technically demanding.

Reviewer #2:

Remarks to the Author:

This manuscript by Lee et al addresses the nature of the elusive flow-transported molecule thought to establish morphological LR asymmetry downstream of ciliary activity in the left-right organizer (LRO); they identify R-Spondin 2 (RSPO2) as a potential left-transported signal.

The authors first show, through in vivo assays in Xenopus embryos, that rspo2, although expressed bilaterally in the LRO, functions LR asymmetrically to determine organ situs and the LR asymmetrical expression patterns of Pitx2 and Dand5. Interestingly, rspo2 can rescue laterality defects caused by

cilia perturbations, suggesting *rspo2* could be a flow-transported morphogen.

In other contexts, RSPO2 acts as both a WNT agonist and a BMP antagonist, yet the authors present several lines of *in vivo* and *in vitro* experiments — (e.g., testing the function of RSPO2 mutants lacking functional FU1, FU2, TSP1 domains required for WNT or BMP transduction, assessing the effect of WNT/BMP misexpression on RSPO2-mediated FGF19 signaling) — suggesting that RSPO2 acts independently of both WNT and BMP signaling in this context. These data indicated that the observed asymmetrical function of *Rspo2* in the *Xenopus* LRO likely occurs via an alternative pathway such as FGF. Indeed, misexpressing *Fgfr4* activity in the *Xenopus* gastrocoel cavity disrupts the LR asymmetry of *Pitx2/Dand5* expression, suggesting that, like *Rspo2*, *Fgfr4* also functions LR asymmetrically at the LRO (but as a right side signal). Moreover, reporter assays and Western blotting in cell culture (and *in vivo*) show that RSPO2 can antagonize FGF19 signaling, and that this antagonism requires FGFR4 activity.

By testing receptor-ligand interactions in cell culture, the study then shows that RSPO2 binding to FGFR4 requires the TSP1 domain of RSPO2. Subsequent biotinylation and endocytosis assays confirmed that RSPO2 reduces cell surface FGFR4 (via LGR4/5, i.e., the FU2 domain) and that the molecules colocalize with the early endosome marker EEA1 and clathrin; thus, RSPO2 may specifically regulate FGFR4 internalization. Consistent with RSPO2 also engaging the E3 ubiquitin ligase ZNRF3 (via its FU1 domain), the authors then show that ZNRF3 promotes FGF19 stimulation, mediates FGFR4 endocytosis in cell culture, and is required for normal *Pitx2* expression *in vivo*. Finally, the study identifies overlapping 10-mer peptides spanning the RSPO2 TSP1 domain that appear to mediate RSPO2 direct binding to FGFR4 *in vitro* and that can disrupt FGF signaling and LR gene (e.g., *dand5*) expression *in vivo*.

The authors conclude that cilia-mediated leftward flow of RSPO2 protein across the LRO causes binding and internalization of FGFR4, thereby eliciting derepression of FGF-mediated right side signaling on the left side of the LRO—thus initiating the Nodal/*Pitx2* cascade on the left. Overall, I found the evidence for the molecular tri-functionality of RSPO2 as a multimodal switch regulating WNT/BMP/FGF signaling to be comprehensive and largely convincing. This work has potentially highly significant implications for the establishment of LR asymmetry although, as the authors themselves acknowledge, the validity of the overall model suffers from a lack of demonstration of asymmetrical accumulation of RSPO2 protein and/or FGF signaling in the LRO.

Major Concerns:

1) In the methods, the authors state that “Intestinal looping was determined by directional rotation of the gut...Abnormal intestinal looping was determined by inverted rotation or loss of looping.” However, in Figure 1 d/e, the digestive organ morphology is incorrectly identified. First, the arrows under “situs anomalies” (indicating the “direction of intestine looping” according to the figure legend) delineate the tadpole’s stomach, not its intestine. Second, the arrows in the different images are not equivalent, i.e., the arrow drawn in the first example provided under “situs anomalies” traverses the stomach tube from its most anterior/proximal end to its most posterior/distal end, but the arrows drawn in the *situs solitus* embryo, as well as the second (middle) example under “situs anomalies”, point from distal to proximal (this mislabeling make the first phenotype appear to be reversed when it is not). Regardless of the annotation, interpreting the phenotypes shown as “situs anomalies” is somewhat misleading since, in all cases, the proximal intestine loop (i.e., the “coil origin”, see *Developmental Biology* 2000: 223, 291–306) is still properly oriented (on the embryo’s left) even though the stomach and intestine are variably shortened. Since RSPO2-related pathways (Wnts/BMPs) also play roles in gut morphogenesis, the phenotypes shown may simply reflect organ-specific morphogenesis defects, rather than situs anomalies per se. It is therefore important to know the extent to which both heart and gut LR asymmetries are affected by *rspo2* misexpression, i.e., by separating out heart vs gut situs defects in Fig 1f and 1g. (Dissecting open the visceral cavity and following the path of the GI tract from anterior to posterior provides the clearest confirmation of stomach/intestinal looping phenotypes.)

2) All Western blots should be quantified, both in the main and supplementary figures, throughout the manuscript.

3) Fig 5o: The *znrf3*-MO injected embryos look slightly microcephalic, suggesting possible gastrulation or dorsoanterior body axis deficiencies (which, as the authors are aware, can affect LR asymmetry independently of gene-specific manipulation). Both the microcephaly and *Pitx2* asymmetry are rescued by co-injecting wt *znrf3* mRNA, but the methods do not specify whether the *znrf3* MO rescues were performed with MO-resistant *znrf3* mRNA; this is crucial for proper interpretation of rescue.

4) The paper does not show that *Rspo2* protein is actually distributed in a concentration gradient across the LRO, nor do they show that FGF signaling is LR asymmetrical. As these are critical features of their model, they should show LR asymmetric accumulation and/or binding of fluorescent *RSPO2*, and/or visualize LR asymmetric activation and/or internalization of *FGFR4*, in explants of the LRO. In addition to this, the paper should at least show (by in situ hybridization or IHC) exactly where other *RSPOs*, *FGF19*, *FGFR1/4*, *ZNRF3*, and *LGR4/5* are expressed in the GRP region during flow stages.

Minor concerns:

1) Fig 2i: Adding "L" and "R", e.g., as in Fig 3M, would improve clarity.

2) What evidence confirms that recombinant *Wnt3A* and *Dkk* proteins are actually active in the gastrocoel cavity (i.e., what were the positive controls or activity assays for Fig 3A,B)? Also, what was the negative control for injecting exogenous protein into the gastrocoel cavity (Fig 3C)?

3) Fig 3e/h. There appears to be a difference between the left and right side expression of *pitx2* in the eye in the BLU9931/AP20187 injected embryos. Can the authors provide an explanation for this craniofacial asymmetry?

4) The term "anti-FGF-disrupting" (p. 17, line 304) is confusing. Is this meant to suggest that the TK peptide has anti-FGF signaling activity, or that it prevents FGF disruption, or...?

Point-by-point response to the reviewers' comments

We appreciate the constructive comments of the reviewers, which provided valuable guidance to improve our manuscript. We have now addressed all of the comments, including by **17 new main- and 24 new supplementary figure panels**. Importantly, we now reveal that **(1) FGF signaling forms a gradient (Reviewer #1 and #2)** and that **(2) this gradient requires Rspo2 (Reviewer #1) as well as ciliary flow**. We believe these exciting findings significantly strengthen our study.

Reviewer 1:

Lee et al found that Rspo2 is expressed bilaterally at the Xenopus LRO, and examined its role in L-R asymmetry by Mo injection. Injection of Rspo2 Mo into the left side of the embryos showed more profound effects than the injection to the right side, suggesting that Rspo2 functions mainly on the left-side of the LRO. Since bilateral activation of FGF4 signaling with ifgfrb and AP20187 induced bilateral Dnd5 expression, and this effect was partially suppressed by excess of Rspo2, it was proposed that Rspo2 inhibits FGF4 signaling. Molecular mechanism of Rspo2 action was examined with cultured cells. Those data obtained with cultured cells suggested that i) Rspo2 interacts with FGFR4 protein and inhibits FGF signaling, 2) Rspo2 stimulates the internalization of FGFR4, antagonizing FGF signaling. By combining the data obtained with Xenopus embryos and those with cultured cells, the authors proposed that Rspo2 generates asymmetry in FGF signaling at the LRO. The paper contains several new interesting findings, such as a new mechanism of Rspo2 action. However, some of the key data obtained with cultured cells must be tested with embryos whenever feasible, if the authors wish to propose a new LR mechanism in embryos.

We further validated anti-FGF signaling mechanism in *Xenopus* in **New Fig. 7, New Supplementary Fig. 5, and New Supplementary Fig. 7** (see below).

Comment 1. *Most importantly, the level of FGF signaling at the LRO of frog embryos must be examined, such as with a pERK antibody, to see if FGF signaling is indeed asymmetric at the LRO in non-treated (wild-type) Xenopus embryos.*

We now performed immunofluorescence microscopy (IF) and western blot analyses in flow-stage LRO explants to analyze pERK levels. As the reviewer envisaged, both clearly reveal an up to ~6-fold FGF signaling gradient in the LRO from right- to left side with **(1) ciliary flow dependency**,

(2) *Rspo2* dependency, and (3) *Fgfr4* dependency (**New Fig. 7a-h; New Supplementary Fig. 7a-k**). This LR asymmetry is significantly disrupted in flow compromised LRO, corroborating that ciliary flow as a causative for the FGF signaling gradient (**New Fig. 7e, f**). Moreover, pre-flow stage LRO shows bilaterally symmetric pERK1 (**New Fig. 7a, b**).

These exciting findings strongly corroborate our model of *Rspo2* as the Hirokawa morphogen.

If FGF signaling is asymmetric, is its asymmetry dependent on Rspo2? This would be the most essential data that is required for the model presented in this paper.

Yes, *Rspo2* is required to generate the FGF signaling gradient. Overexpression *RSPO2* during leftward flow drastically lowers pERK1 levels and flattens the FGF signaling gradient, confirming anti-FGF function of *RSPO2* (**New Fig. 7c, d; New Supplementary Fig. 7b; New Supplementary Fig. 7e-g**). Conversely, knockdown of *rspo2* increases pERK1 levels at the left side LRO, thereby equalizing the FGF signaling gradient (**New Fig. 7g, h; New Supplementary Fig. 7d; New Supplementary Fig. 7h-j**).

Lastly, we confirm that it is the *RSPO2-FGFR4* antagonism that generates the FGF signaling gradient: Pharmacological inhibition of *Fgfr4* disrupts the pERK1 gradient similar to gain of *RSPO2*, while an *Rspo2-Fgfr4* intervening peptide equalizes the gradient similar to *rspo2* knockdown (**New Fig. 7e, f; New Supplementary Fig. 7c; New Supplementary Fig. 7k**).

Based on these critical advances made during the revision, we now updated our model including an FGF signaling gradient in sinistro-dextral axis formation (**New Fig. 7i; New Fig. 8a**) and elaborated this finding in The Discussion:

“...iii) *Rspo2* establishes a sinistro-dextral FGF/ERK signaling gradient in the LRO. Thus, similar to the Spemann-Mangold organizer, where BMP- and Wnt morphogen gradients are generated via secreted antagonists such as Chordin and Dkk1, *Rspo2* is not an instructive signal but instead it restricts the dextralizing function of FGF signaling.”

Comment 2. *Internalization of FGFR4 by Rspo2 (Fig. 5g-l), which is another important mechanism proposed by this paper, can be examined with frog embryos.*

Since the anti-FGFR4 antibody used in the MS fails to detect endogenous *Xenopus Fgfr4*, we now injected *fgfr4-EYFP* mRNA. To bypass endogenous *Rspo2* function, we utilized *Xenopus*

animal cap explants where *rspo2* is barely expressed. Upon injection, *rspo2* WT but not *rspo2*^{ΔTSP1} reduces cell surface Fgfr4 levels (**New Supplementary Fig. 5e-g**), corroborating internalization of Fgfr4 by Rspo2. Consistent with the mechanism in human cells, *rspo2*-mediated reduction of cell surface Fgfr4 requires Znr3, specifically its E3 ubiquitin ligase function (**New Supplementary Fig. 5h-j**). Unlike in human cells, we found no increase of vesicular Fgfr4 upon *rspo2* overexpression but instead a drastic reduction of overall Fgfr4 levels, suggesting lysosomal degradation (**New Supplementary Fig. 5e-j**).

Collectively, we conclude that Rspo2 removes Fgfr4 from the cell surface in a Znr3-dependent manner not only in human cultured cells but also in *Xenopus* embryonic cells.

Also, is the level of the extra-cellular domain of FGFR4 on the cell surface reduced by Rspo2? This can be investigated with an antibody against the extra-cellular domain.

We tested a new antibody against the extracellular domain (ECD) of FGFR4, but it did not show specific signals in either IF or western blot analyses using human cells and *Xenopus* explants. However, our FGFR4 endocytosis assay (**Fig. 5f**) already shows that FGFR4^{ECD} is captured by RSPO2 and reduced from the cell surface.

We now indicate this point clearly in the MS:

“Cell surface proteins were labeled **extracellularly** with a biotin-coupled cross-linker containing a disulfide bridge, which can be cleaved-off by the reducing agent MesNa. At t_0 , **extracellular domain labeled FGFR4** was quantitatively removed by MesNa (**Fig. 5f**), i.e. no FGFR4 was internalized.”

Comment 3. *The authors propose that Rspo2 protein is transported toward the left side of LRO by the flow (Fig. 3m, and Discussion). If Rspo2 protein is transported from the right to the left side, why does *rspo2* Mo show more profound effects on the left-side?*

The simplest explanation is that in left-side Mo injection, ciliary flow fails to transport enough Rspo2 from the right side to the left to compensate for the complete loss of *rspo2* from the very side where it matters. In other words, left-side Mo injection works against the flow, while right-side Mo injection works with the flow.

Comment 4. *Regulation of Dand5 mRNA at the LRO of Xenopus embryo is post-transcriptional (Maerker et al 2021)¹. Does FGF signaling stabilize Dand5 mRNA, in cultured cells or in embryos?*

To address this question, we now injected recombinant human FGF19 protein as well as FGFR4 inhibitor BLU9931 and examined *dand5* expression in flow-stage LRO. FGF19-injected embryos display symmetric *dand5* expression similar to *Dicer* morphants¹, while BLU9931 abolishes *dand5* expression (**New Supplementary Fig. 8a-d**). These results suggest that FGF19-FGFR4 signaling prevents post-transcriptional *dand5* mRNA decay during LR symmetry breakage. Further elucidation of the dextralizing pathway will be interesting for future studies.

Comment 5. *It has been shown by others (Schneider et al, 2019)² that inhibition of FGF signaling by Sprouty impairs specification of the LRO cells (somitic GRP cells), down-regulating the level of Dand5 (and Nodal) mRNA at the LRO of Xenopus embryo. This discrepancy needs to be clarified.*

We do not think that our data are discrepant to Schneider *et al*², since stage- and mode-of-actions between Sprouty and Rspo2 are very different. Specifically, Sprouty acts as i) FGF/Ca²⁺ signaling antagonist and ii) acts prior to leftward flow, i.e. late gastrulation, and is required for formation of somitic GRP cells, which function as LRO flow sensor². Different from Sprouty, Rspo2 acts as i) FGF/ERK signaling antagonist and ii) during flow stages and is required to generate the FGF signaling gradient as now confirmed in **New Fig. 7**.

Comment 6. *Whether Rspo2 protein is indeed transported by the flow is the next issue because it is technically demanding.*

We thank the reviewer for acknowledging the technical challenges to visualize Rspo2 protein in the LRO.

Reviewer 2:

This manuscript by Lee et al addresses the nature of the elusive flow-transported molecule thought to establish morphological LR asymmetry downstream of ciliary activity in the left-right organizer (LRO); they identify R-Spondin 2 (RSPO2) as a potential left-transported signal.

The authors first show, through in vivo assays in Xenopus embryos, that rspo2, although expressed bilaterally in the LRO, functions LR asymmetrically to determine organ situs and the LR asymmetrical expression patterns of Pitx2 and Dand5. Interestingly, rspo2 can rescue laterality defects caused by cilia perturbations, suggesting rspo2 could be a flow-transported morphogen.

In other contexts, RSPO2 acts as both a WNT agonist and a BMP antagonist, yet the authors present several lines of in vivo and in vitro experiments — (e.g., testing the function of RSPO2 mutants lacking functional FU1, FU2, TSP1 domains required for WNT or BMP transduction, assessing the effect of WNT/BMP misexpression on RSPO2-mediated FGF19 signaling) — suggesting that RSPO2 acts independently of both WNT and BMP signaling in this context. These data indicated that the observed asymmetrical function of Rspo2 in the Xenopus LRO likely occurs via an alternative pathway such as FGF. Indeed, misexpressing Fgfr4 activity in the Xenopus gastrocoel cavity disrupts the LR asymmetry of Pitx2/Dand5 expression, suggesting that, like Rspo2, Fgfr4 also functions LR asymmetrically at the LRO (but as a right side signal). Moreover, reporter assays and Western blotting in cell culture (and in vivo) show that RSPO2 can antagonize FGF19 signaling, and that this antagonism requires FGFR4 activity.

By testing receptor-ligand interactions in cell culture, the study then shows that RSPO2 binding to FGFR4 requires the TSP1 domain of RSPO2. Subsequent biotinylation and endocytosis assays confirmed that RSPO2 reduces cell surface FGFR4 (via LGR4/5, i.e., the FU2 domain) and that the molecules colocalize with the early endosome marker EEA1 and clathrin; thus, RSPO2 may specifically regulate FGFR4 internalization. Consistent with RSPO2 also engaging the E3 ubiquitin ligase ZNRF3 (via its FU1 domain), the authors then show that ZNRF3 promotes FGF19 stimulation, mediates FGFR4 endocytosis in cell culture, and is required for normal Pitx2 expression in vivo. Finally, the study identifies overlapping 10-mer peptides spanning the RSPO2 TSP1 domain that appear to mediate RSPO2 direct binding to FGFR4 in vitro and that can disrupt FGF signaling and LR gene (e.g., dand5) expression in vivo.

The authors conclude that cilia-mediated leftward flow of RSPO2 protein across the LRO causes binding and internalization of FGFR4, thereby eliciting derepression of FGF-mediated right side

signaling on the left side of the LRO—thus initiating the Nodal/Pitx2 cascade on the left. Overall, I found the evidence for the molecular tri-functionality of RSPO2 as a multimodal switch regulating WNT/BMP/FGF signaling to be comprehensive and largely convincing. This work has potentially highly significant implications for the establishment of LR asymmetry although, as the authors themselves acknowledge, the validity of the overall model suffers from a lack of demonstration of asymmetrical accumulation of RSPO2 protein and/or FGF signaling in the LRO.

We thank the reviewer for the thorough read of our MS and pointing out lack of the analyses for asymmetric FGF signaling in the LRO. We now demonstrate LR asymmetric FGF signaling at the LRO, which validates our model.

Major points

Comment 1. *In the methods, the authors state that “Intestinal looping was determined by directional rotation of the gut...Abnormal intestinal looping was determined by inverted rotation or loss of looping.” However, in Figure 1 d/e, the digestive organ morphology is incorrectly identified. First, the arrows under “situs anomalies” (indicating the “direction of intestine looping” according to the figure legend) delineate the tadpole’s stomach, not its intestine. Second, the arrows in the different images are not equivalent, i.e., the arrow drawn in the first example provided under “situs anomalies” traverses the stomach tube from its most anterior/proximal end to its most posterior/distal end, but the arrows drawn in the situs solitus embryo, as well as the second (middle) example under “situs anomalies”, point from distal to proximal (this mislabeling make the first phenotype appear to be reversed when it is not). Regardless of the annotation, interpreting the phenotypes shown as “situs anomalies” is somewhat misleading since, in all cases, the proximal intestine loop (i.e., the “coil origin”, see *Developmental Biology* 2000: 223, 291–306) is still properly oriented (on the embryo’s left) even though the stomach and intestine are variably shortened. Since RSPO2-related pathways (Wnts/BMPs) also play roles in gut morphogenesis, the phenotypes shown may simply reflect organ-specific morphogenesis defects, rather than situs anomalies per se. It is therefore important to know the extent to which both heart and gut LR asymmetries are affected by *rspo2* misexpression, i.e., by separating out heart vs gut situs defects in Fig 1f and 1g. (Dissecting open the visceral cavity and following the path of the GI tract from anterior to posterior provides the clearest confirmation of stomach/intestinal looping phenotypes.)*

We appreciate the reviewer’s constructive comment to amend our incorrect interpretation for organ *situs*. As suggested, we now examined separately heart looping and gut looping by

dissection from St. 42 embryos (**New Fig. 1e-j**). EU law prohibits *Xenopus* experiments without separate (and very time-consuming) license beyond St. 43, hence we were not able to investigate gut coiling origin and direction at stages that are typical readouts in *Xenopus* LR asymmetry studies³.

Nevertheless, re-analyses of St. 42 embryos with *rspo2* overexpression or *rspo2* knockdown shows reversed heart looping (**New Fig. 1e-g**), corroborating the previous Figure 1f. As for gut looping, LR asymmetry is already evident at St. 42, marked by formation of the left-side specific 'pancreatic bay' concavity of the gastroduodenal loop⁴. *Rspo2* overexpression embryos display gut *situs* defects (**New Fig. 1h-j; Supplementary Fig. 1b**), which phenocopies *pitx2* overexpression⁴. Of note, as the reviewer pointed out, *rspo2* overexpression again induced shortened gut but this is also reported in *pitx2* overexpressing embryos⁵.

Overall, our re-evaluation of *rspo2* GOF and LOF phenotypes clearly demonstrates that *Rspo2* misregulation induces anomalies of both heart- and gut *situs*, rather than heart- or gut-specific morphogenesis. We have exchanged **previous Fig. 1e-h** for **New Fig. 1e-j**.

Comment 2. *All Western blots should be quantified, both in the main and supplementary figures, throughout the manuscript.*

We now provide the quantification of all western blot analyses in the MS.

Comment 3. *Fig 5o: The znrf3-MO injected embryos look slightly microcephalic, suggesting possible gastrulation or dorsoanterior body axis deficiencies (which, as the authors are aware, can affect LR asymmetry independently of gene-specific manipulation). Both the microcephaly and Pitx2 asymmetry are rescued by co-injecting wt znrf3 mRNA, but the methods do not specify whether the znrf3 MO rescues were performed with MO-resistant znrf3 mRNA; this is crucial for proper interpretation of rescue.*

We apologize for the missing information. We had injected a MO-resistant human ZNRF3 mRNA, which was already specificity-validated⁶. The info was added to the MS.

Comment 4. *The paper does not show that Rspo2 protein is actually distributed in a concentration gradient across the LRO, nor do they show that FGF signaling is LR asymmetrical. As these are*

critical features of their model, they should show LR asymmetric accumulation and/or binding of fluorescent RSPO2, and/or visualize LR asymmetric activation and/or internalization of FGFR4, in explants of the LRO.

Thank you for bringing up the question of asymmetric Rspo2 distribution and asymmetric FGF signaling. To visualize LR asymmetric activation of FGF-FGFR4 signaling, we have tried to analyze endogenous Fgfr4 localization as well as Fgfr4 phosphorylation using commercial antibodies. However, these antibodies did not show specific staining in IF or western blot analysis. Hence, we focused on FGF/ERK signaling and analyzed phosphorylated ERK1/2 levels (**New Fig. 7; New Supplementary Fig. 7**). As the reviewer envisaged, IF revealed that phosphorylated ERK1 staining is 6x higher at the right side comparing to the left side of the LRO during leftward flow (**New Fig. 7c-d; New Supplementary Fig. 7b**). We confirmed this finding by western blot analyses (**New Supplementary Fig. 7e-g**). We now updated our model including FGF signaling gradient in sinistro-dextral axis formation (**New Fig. 7i; New Fig. 8a**).

Concerning the visualization of RSPO2, we ask for indulgence that monitoring the protein is technically very challenging and will require a separate study: None of the commercial antibodies is working against *Xenopus* Rspo2. Injecting HRP-tagged RSPO2 recombinant protein was an alternative to visualize RSPO2, giving the most specific and strong signal in human cells (**Fig. 5n**). However, RSPO2-HRP is not applicable to the LRO since the size of RSPO2-HRP is over 70 kDa, which is much bigger than Rspo2 (25 kDa) and the expected size of extracellular proteins transported by ciliary flow (15-50 kDa). Therefore, an option to visualize Rspo2 is generating transgenic *Xenopus* which harbors *rspo2* endogenously tagged with fluorescent proteins. However, generating, characterizing, and imaging transgenic *Xenopus* embryo are way beyond the scope of this paper.

In addition to this, the paper should at least show (by in situ hybridization or IHC) exactly where other RSPOs, FGF19, FGFR1/4, ZNRF3, and LGR4/5 are expressed in the GRP region during flow stages.

We now provide *in situ* hybridization of *rspo1*, *rspo3*, *fgf19*, *fgfr1*, *fgfr4*, *znrf3*, and *lgr4* in flow stage LRO explants (St. 18-19) (**New Supplementary Fig. 1a; New Supplementary Fig. 5a**). Of note, *rspo4* has not been reported in *Xenopus laevis*.

Minor Concerns:

Comment 1. *Fig 2i: Adding “L” and “R”, e.g., as in Fig 3M, would improve clarity.*

We have revised the figure accordingly.

Comment 2. *What evidence confirms that recombinant Wnt3A and Dkk proteins are actually active in the gastrocoel cavity (i.e., what were the positive controls or activity assays for Fig 3A, B)? Also, what was the negative control for injecting exogenous protein into the gastrocoel cavity (Fig 3C)?*

Supplementary Fig. 1g and 1h had confirmed that gastrocoel injection of Wnt3A protein increases activate beta-catenin (dephosphorylated β -catenin), while DKK1 protein decreases it in the LRO tissues. Since functionally validated Wnt3A and DKK1 proteins did not induce *pitx2c* misexpression, these serve as the exogenous protein controls for Fig. 3C.

Comment 3. *Fig 3e/h. There appears to be a difference between the left and right-side expression of *pitx2* in the eye in the BLU9931/AP20187 injected embryos. Can the authors provide an explanation for this craniofacial asymmetry?*

The subtle difference that the reviewer noticed is probably due to a slight developmental delay in craniofacial development in DMSO-injected embryos and that exhibits a LR difference.

Comment 4. *The term “anti-FGF-disrupting” (p. 17, line 304) is confusing. Is this meant to suggest that the TK peptide has anti-FGF signaling activity, or that it prevents FGF disruption, or...?*

We apologize for the confusion. Now we have corrected the term as “TK peptide, which specifically blocks Rspo2-Fgfr4 interaction.”.

References

- 1 Maerker, M. *et al.* Bicc1 and Dicer regulate left-right patterning through post-transcriptional control of the Nodal inhibitor Dand5. *Nat Commun* **12** (2021).
- 2 Schneider, I., Kreis, J., Schweickert, A., Blum, M. & Vick, P. A dual function of FGF signaling in Xenopus left-right axis formation. *Development* **146** (2019).
- 3 Branford, W. W., Essner, J. J. & Yost, H. J. Regulation of gut and heart left-right asymmetry by context-dependent interactions between xenopus lefty and BMP4 signaling. *Dev Biol* **223** (2000).
- 4 Muller, J. K., Prather, D. R. & Nascone-Yoder, N. M. Left-right asymmetric morphogenesis in the Xenopus digestive system. *Dev Dyn* **228** (2003).
- 5 Campione, M. *et al.* The homeobox gene Pitx2: mediator of asymmetric left-right signaling in vertebrate heart and gut looping. *Development* **126** (1999).
- 6 Chang, L. S., Kim, M., Glinka, A., Reinhard, C. & Niehrs, C. The tumor suppressor PTPRK promotes ZNRF3 internalization and is required for Wnt inhibition in the Spemann organizer. *Elife* **9** (2020).

Reviewers' Comments:

Reviewer #1:

Remarks to the Author:

The authors have provided additional data that largely satisfy my concerns. In particular, asymmetric distribution of pERK supports their conclusions. I only have minor comments.

1. The distribution of pERK level at the LRO was quantified along the L-R axis (Fig. 7b d,f,h). The distribution pattern looks different at the anterior and posterior regions of LRO. In Fig. 7b,d,f,h, did authors measure the pERK level in the whole LRO or at a specific level of the LRO. If the latter is the case, the level must be indicated in the images of Figure 7.
2. The word "a gradient " is used repeatedly. This is probably an overstatement, and "asymmetric distribution" would be more appropriate for pERK data, although I would not insist.

Reviewer #2:

Remarks to the Author:

The authors have addressed my concerns with additional explanation or new data. Overall, this is an exciting manuscript with important implications for our understanding of vertebrate left-right axis establishment and propagation.

Point-by-point response to the reviewers' comments

Reviewer #1 (Remarks to the Author):

The authors have provided additional data that largely satisfy my concerns. In particular, asymmetric distribution of pERK supports their conclusions. I only have minor comments.

Thank you.

1. The distribution of pERK level at the LRO was quantified along the L-R axis (Fig. 7b d,f,h). The distribution pattern looks different at the anterior and posterior regions of LRO.

We think it is due to the posterior-enriched FGF19 expression (Supplementary figure 5a). Of note, we found the difference of pERK level along the A-P axis is much smaller (max. 2.3-fold) than the L-R axis (6-fold) in the LRO.

In Fig. 7b,d,f,h, did authors measure the pERK level in the whole LRO or at a specific level of the LRO. If the latter is the case, the level must be indicated in the images of Figure 7.

We measured the pERK level in the whole LRO along the L-R axis.

2. The word “a gradient “ is used repeatedly. This is probably an overstatement, and “asymmetric distribution” would be more appropriate for pERK data, although I would not insist.

We prefer to keep ‘FGF signaling gradient’ throughout the ms, since quantification of the IF indeed revealed pERK slope similar to DV or AP morphogen gradient.

Reviewer #2 (Remarks to the Author):

The authors have addressed my concerns with additional explanation or new data. Overall, this is an exciting manuscript with important implications for our understanding of vertebrate left-right axis establishment and propagation.

Thank you.